# Imitation Learning from a Single Temporally Misaligned Video

**William Huey** [* 1]  **Huaxiaoyue Wang** [* 1]  **Anne Wu** [1]  **Yoav Artzi** [1]  **Sanjiban Choudhury** [1]

## Abstract

We examine the problem of learning sequential tasks from a single visual demonstration. A key challenge arises when demonstrations are *temporally misaligned* due to variations in timing, differences in embodiment, or inconsistencies in execution. Existing approaches treat imitation as a distribution-matching problem, aligning individual frames between the agent and the demonstration. However, we show that such frame-level matching fails to enforce temporal ordering or ensure consistent progress. Our key insight is that matching should instead be defined at the level of sequences. We propose that perfect matching occurs when one sequence successfully covers all the subgoals in the same order as the other sequence. We present ORCA (ORdered Coverage Alignment), a dense per-timestep reward function that measures the probability of the agent covering demonstration frames in the correct order. On temporally misaligned demonstrations, we show that agents trained with the ORCA reward achieve 4.5x improvement ($0.11 \rightarrow 0.50$ average normalized returns) for Meta-world tasks and 6.6x improvement ($6.55 \rightarrow 43.3$ average returns) for Humanoid-v4 tasks compared to the best frame-level matching algorithms. We also provide empirical analysis showing that ORCA is robust to varying levels of temporal misalignment. The project website is at https://portal-cornell.github.io/orca/

## 1. Introduction

Designing reward functions for reinforcement learning (RL) is tedious (Eschmann, 2021), especially for tasks that require completing subgoals in a strict order. A more scalable way is to learn rewards from a video demonstration. However,

learning from videos is challenging because demonstrations are often *temporally misaligned*—variations in timing, embodiment, or execution mean that frame-level matching fails to enforce correct sequencing. To follow a demonstration, an agent must not only match states but also make consistent progress through the subgoals in the right order.

Inverse reinforcement learning (IRL) (Abbeel & Ng, 2004; Ziebart et al., 2008) provides a principled way to infer rewards by matching the learner's trajectory to expert demonstrations. Recent IRL approaches apply optimal transport (OT) (Peyré & Cuturi, 2020) to align visual embeddings between frames (Cohen et al., 2022; Fu et al., 2024c; Guzey et al., 2024; Haldar et al., 2023a;b; Liu et al., 2024; Tian et al., 2024). However, these methods operate at the frame level, ignoring temporal dependencies: an agent can revisit subgoals, skip ahead, or stall without penalty. As a result, frame-matching approaches fail on sequence-matching tasks, where the correct subgoal order is crucial for success.

Our key insight is that ***matching should be defined at the sequence level instead of the frame level*** when learning rewards for sequence-matching tasks. Under temporal misalignment, we claim that two sequences only match if one sequence aligns with all of the other sequence's subgoals in the same order. This leads to an intuitive and principled reward function for sequence-matching tasks: an agent should be rewarded for covering all subgoals in order, without requiring precise timing alignment.

We propose ORCA (ORdered Coverage Alignment), which computes dense rewards given a single video demonstration and a visual learner trajectory, as shown in Fig. 1. Concretely, the reward function is recursively defined as the probability that (1) the learner currently occupies the subgoal specified by the final video frame, and (2) the learner has already covered all prior frames in the correct order. In practice, we train an ORCA policy via RL in two stages. First, we initialize the policy by training with rewards that assume a temporally aligned demonstration. Second, we refine this policy with ORCA rewards to significantly improve its efficiency and performance. Our key contributions are:

1. A novel, principled reward function class ORCA that formulates the reward as an ordered coverage problem.

2. Analysis on the weakness of rewards based on optimal transport and other frame-level matching algorithms.

---
[*]Equal contribution [1]Cornell University. Correspondence to: William Huey <wph52@cornell.edu>, Huaxiaoyue (Yuki) Wang <yukiwang@cs.cornell.edu>.

*Proceedings of the 42nd International Conference on Machine Learning*, Vancouver, Canada. PMLR 267, 2025. Copyright 2025 by the author(s).

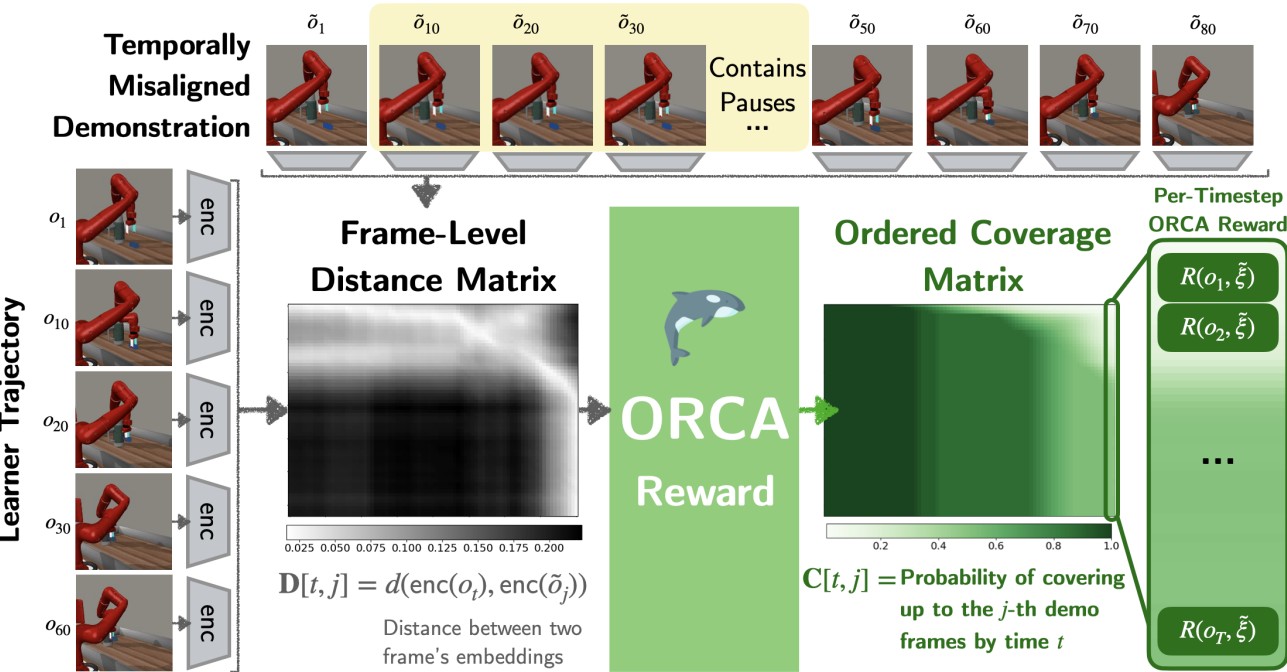

*Figure 1.* **ORCA overview.** The expert video demonstrates the *Stick-Push* task, where the robot must grasp the tool before pushing the water bottle. However, this demonstration is *temporally misaligned* because it contains long pauses before picking up the tool. To learn from this demonstration, ORCA provides a per-timestep reward for the visual learner trajectory $\xi = \{o_t\}_{t=1}^T$. (1) Each frame is passed through an off-the-shelf visual encoder before calculating its distance with respect to the other frames. (2) The ORCA reward calculates the probability that the learner has covered *all* the frames in the correct order.

3. Experiments showing that the ORCA reward can effectively and efficiently train RL agents to achieve 4.5x improvement ($0.11 \rightarrow 0.50$ average normalized return) for Meta-world tasks and 6.6x improvement ($6.55 \rightarrow 43.3$ average return) for `Humanoid-v4` tasks compared to the best frame-level matching approach.

## 2. Problem Formulation

Sequence-matching problems focus on tasks where it is critical to follow the entire sequence in the correct order: e.g., for assembly tasks, robots must assemble all parts in the demonstrated order. We model the problem as a Markov Decision Process (MDP) $(\mathcal{S}, \mathcal{A}, \mathcal{T}, r, \gamma)$. At time $t$, the agent at state $s_t \in \mathcal{S}$ receives an *image* observation $o_t \in \mathcal{O}$ of the state. On taking action $a_t \in \mathcal{A}$, it transitions to a new state $s_{t+1} \in \mathcal{S}$ and gets a new observation $o_{t+1} \in \mathcal{O}$.

*We assume the reward function for this task is unknown.* Instead, we assume access to a *single* visual demonstration of the task, $\tilde{\xi} = \{\tilde{o}_1, \ldots, \tilde{o}_{\tilde{T}}\}$. The robot policy must follow all subgoals in the same order as the demonstration. Importantly, this demonstration may be *temporally misaligned* with the learner's trajectory and have a different execution speed. Additionally, the demonstration lacks state and action labels, rendering classical imitation learning inapplicable.

Instead, we approach this as an inverse reinforcement learning (IRL) problem, where the reward $\mathcal{R}(o_{1:t}, \tilde{\xi})$ is defined as a function of the visual demonstration $\tilde{\xi}$ and the learner's observations. The goal is to learn a policy $\pi^*(a|s)$ that maximizes the expected discounted sum of rewards:

$$\pi^* = \arg\max_\pi \mathbb{E}_\pi \left[ \sum_{t=1}^T \gamma^t \mathcal{R}(o_{1:t}, \tilde{\xi}) \right].$$

## 3. Desiderata of Sequence-Matching Rewards

IRL can be formulated as a distribution matching problem between the learner and the demonstrator on "moments", i.e., expectations of reward basis functions (Swamy et al., 2021). Hence, solving IRL is equivalent to optimizing Integral Probability Metrics (IPMs) between the learner and demonstrator distributions (Sun et al., 2019). Recent works use optimal transport (OT) by minimizing the Wasserstein distance (an IPM) between embeddings of the frames in the learner and demonstration trajectories. They work with Markovian moments that depend on only a single frame.

However, *the true reward function for a sequence-matching task depends on trajectory history* to determine whether the agent has followed the subgoals in order, so it is not in the span of Markovian moments. Instead of matching the distri-

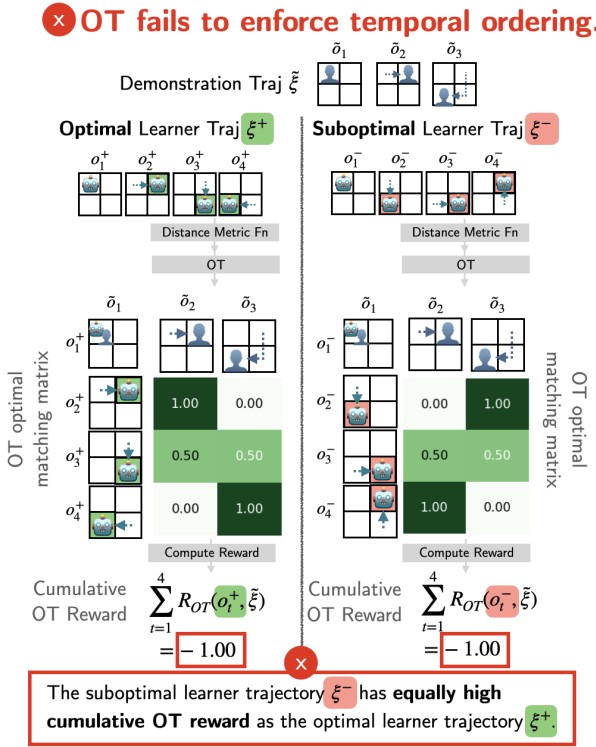

**OT fails to enforce temporal ordering.**

*Figure 2.* **Failure cases for OT reward.** The suboptimal learner trajectory $\xi^-$ moves in the counter-clockwise direction while optimal one $\xi^+$ moves clockwise. Unlike the learner, the agent in the demonstration trajectory can move multiple cells per timestep.

bution over frames, we should be matching the distribution over trajectories. We propose that the corresponding reward should be a measure of:

1. **Subgoal Ordering**: A learner trajectory that completes the subgoals in the correct order should receive a higher cumulative reward than one that completes the subgoals in the wrong order.
2. **Subgoal Coverage**: A learner trajectory that completes more of the subgoals should receive a higher cumulative reward than one that completes less of them.

Rewards that use frame-level matching fail to satisfy these desiderata: OT fails to enforce subgoal covering (Sec. 3.1), and mitigations to OT's problem fail to enforce full subgoal coverage (Sec. 3.2 and 3.3).

### 3.1. Optimal Transport Fails to Respect Ordering

Prior works in imitation learning use Optimal Transport (OT) (Peyré & Cuturi, 2020), specifically the Wasserstein distance, to measure how closely a learner trajectory matches a demonstration trajectory (Fu et al., 2024c; Kedia et al., 2025; Papagiannis & Li, 2022; Tian et al., 2024). These approaches assume a Markovian distance

metric $d(\cdot, \cdot)$ that measures the distance between the embeddings of a learner and demonstration frame. Given a learner trajectory $\xi = \{o_t\}_{t=1}^T$ and a video demonstration $\tilde{\xi} = \{\tilde{o}_j\}_{j=1}^{\tilde{T}}$, they define corresponding learner and demonstration distributions that are uniform in time. Specifically, $\rho = \frac{1}{T} \sum_{t=1}^T \delta_{o_t}$, where $\delta_{o_t}$ is a Dirac distribution centered on $o_t$. The demonstration similarly has a distribution of $\tilde{\rho} = \frac{1}{\tilde{T}} \sum_{j=1}^{\tilde{T}} \delta_{\tilde{o}_j}$. Consequently, the matching matrices are evenly weighted, defined as the set $M = \{\mu \in \mathbb{R}^{T \times \tilde{T}} : \mu\mathbf{1} = \frac{1}{T}, \mu^T \mathbf{1} = \frac{1}{\tilde{T}}\}$. With an entropy regularizer $\mathcal{H}(\cdot)$, OT solves for the optimal matching matrix:

$$\mu^* = \arg \min_{\mu \in M} \sum_{t=1}^T \sum_{j=1}^{\tilde{T}} d(o_t, \tilde{o}_j)\mu_{t,j} - \epsilon\mathcal{H}(\mu). \quad (1)$$

The OT reward is:

$$\mathcal{R}_{\text{OT}}(o_t, \xi, \tilde{\xi}) = -\sum_{j=1}^{\tilde{T}} d(o_t, \tilde{o}_j)\mu_{i,j}^*. \quad (2)$$

Although this is a non-Markovian reward function, *it still matches Markovian moments*. This is clear in its failure to enforce subgoal ordering.

**Counterexample 3.1.** *There exists an MDP (Fig. 2) where OT fails to penalize violations of temporal ordering.*

The OT reward uses a Markovian distance based on two frame embeddings, and there are no temporal constraints on $\mu^*$ in (1). Thus, OT does not penalize trajectories that complete subgoals in the wrong order. Fig. 2 shows an example where a suboptimal trajectory that visits the subgoals in the *reversed* order has the same OT reward as the optimal one.

### 3.2. Dynamic Time Warping Fails to Cover All Subgoals

Dynamic Time Warping (DTW) (Sakoe & Chiba, 1978) overcomes the subgoal ordering problem of OT by temporally aligning the learner and demonstration trajectories, while allowing for flexible distribution of the assignment over each demonstration frame. Specifically, each matching matrix $\mu$ must align the beginning and the end of two trajectories (i.e., $\mu_{1,1} = 1$ and $\mu_{T,\tilde{T}} = 1$), and the indices of its positive entries must be non-decreasing in time. Subject to these constraints, DTW solves for the optimal matching:
$\mu^* = \arg \min_\mu \sum_{t=1}^T \sum_{j=1}^{\tilde{T}} \mu_{t,j} d(o_t, \tilde{o}_j)$.

Given $\mu^*$, the DTW reward is the same as (2). DTW's temporal constraints allow it to avoid the example failure of OT (shown in Fig. 7 of Appendix A.2), but they lead to a new failure mode.

**Counterexample 3.2.** *There exists an MDP (Fig. 8, App. A.2) where DTW fails to penalize incomplete subgoal coverage.*

Although DTW constrains the order of assignment, it does not limit the number of learner frames matched with each subgoal. Once the learner reaches an intermediate subgoal, it can achieve high DTW reward by remaining in that subgoal until the last few frames of the trajectory. Fig. 8 of Appendix A.2 shows an example where a trajectory stuck in the second subgoal achieves the same DTW reward as a trajectory that completes all subgoals. This is a local minimum that could cause RL agents to stall at earlier subgoals.

### 3.3. TemporalOT Fails to Cover All Subgoals under Temporal Misalignment

Recognizing the limitations of OT, Fu et al. (2024c) propose TemporalOT, which alters the OT objective to optimize over a temporally masked cost matrix. Specifically, the mask is a variant of the diagonal matrix with a hyperparameter $k_w$ to define the width of the diagonal:

$$W_{t,j} = \begin{cases} 1, & \text{if } j \in [t - k_w, t + k_w], \\ 0, & \text{otherwise.} \end{cases} \qquad (3)$$

Consequently, TemporalOT solves for the optimal matching matrix $\mu^* \in M$ with the same constraints as (1):

$$\mu^* = \arg\min_{\mu \in M} \sum_{t=1}^{T} \sum_{j=1}^{\tilde{T}} W_{t,j} d(o_t, \tilde{o}_j) \mu_{t,j} - \epsilon \mathcal{H}(W \odot \mu). \qquad (4)$$

Given $\mu^*$, the TemporalOT reward is the same as (2).

**Counterexample 3.3.** *There exists an MDP (Fig. 9, App. A.2) where TemporalOT fails to penalize trajectories that do not reach every subgoal given a temporally misaligned demonstration.*

To define the mask window, TemporalOT makes the strong assumption that the demonstration is temporally aligned with the learner trajectory. Regardless of how temporally misaligned a demonstration is, TemporalOT's matching matrix always approximates a diagonal matrix, matching an equal proportion of learner frames to each subgoal. Fig. 9 of Appendix A.2 shows an example where a suboptimal, slower policy that does not reach later subgoals has a higher TemporalOT reward than an optimal policy that makes consistent progress to complete the task.

## 4. Approach

We introduce ORCA (ORdered Coverage Alignment), a reward function that measures, at each time step, the probability that (1) the learner currently occupies the subgoal specified by the final video frame, and (2) it has already covered all prior frames in the correct order. Since the ORCA reward at time $t$ depends on the learner trajectory up until $t$, it models the non-Markovian nature of sequence-matching tasks. In Sec. 4.2, we theoretically analyze and prove that ORCA satisfies the desiderata proposed in Sec. 3.

### 4.1. ORCA: Ordered Coverage Reward Function

We define the ordered coverage for two sequences as the probability that one sequence covers all the subgoals in the same order as the other sequence. Ordered coverage shares the same recursive nature as the true reward function of sequence-matching tasks: it depends on how well the final subgoal is covered and how well previous subgoals have already been covered. We next present how to calculate ordered coverage and use it to compute rewards.

**Calculate ordered coverage via dynamic programming.** Given a video demonstration and a learner trajectory, we can calculate ordered coverage between the two sequences by computing the matrix $C_{t,j}$, which is the probability that the segment of the learner trajectory from time 1 to $t$ has covered the first to the $j$-th subgoals in the correct order. By the recursive definition of ordered coverage, $C_{t,j}$ can be expressed with two components: (1) the ordered coverage until the $(j-1)$-th subgoal and (2) how well the learner is currently occupying the $j$-th subgoal. In addition, ordered coverage should be nondecreasing over time: i.e., once the learner has covered a subgoal at a timestep, it is always at least equally successful in future timesteps. Let $G_{t,j}$ denote the event that the learner occupies the $j$-th demonstration subgoal at timestep $t$. We assume that the probability for this event is proportional to the negative exponent distance: $P(G_{t,j}) \propto \exp(-\lambda d(o_t, \tilde{o}_j))$, where $\lambda$ is a temperature hyperparameter. For simplicity, we substitute $P(G_{t,j})$ with $P_{t,j}$. Thus, we recursively calculate ordered coverage:

$$C_{t,j} = \max\{C_{t-1,j}, C_{t,j-1} P_{t,j}\}. \qquad (5)$$

Algorithm 1 shows the entire computation.

**ORCA reward function.** In most robotics tasks, the learner should stay in the final demonstration subgoal after covering all previous subgoals. A reward function that directly uses the ordered coverage for the final subgoal does not satisfy this requirement: even if the learner does not remain occupying the final subgoal, as long as it has covered the subgoal at one timestep, the maximum operator in (5) keeps the reward high for remaining timesteps. Instead of simply equating the reward to $C_{t,\tilde{T}}$, the final ORCA reward at timestep $t$ is:

$$\mathcal{R}_{\text{ORCA}}(o_t, \xi, \tilde{\xi}) = C_{t,\tilde{T}-1} P_{t,\tilde{T}}. \qquad (6)$$

**Runtime complexity.** ORCA's runtime complexity is $\mathcal{O}(T \cdot \tilde{T})$. This is the lower bound time complexity of any method that relies on a frame-level distance matrix, including all OT-based methods. To validate this, we experimentally tested the latency of ORCA against a variety of common baselines, finding that ORCA is faster than TemporalOT and comparable to OT. See Appendix C.6 for details.

**Algorithm 1** `ORCA` Rewards.

---

**Input:** learner traj $\xi = \{o_t\}_{t=1}^{T}$, demo traj $\tilde{\xi} = \{\tilde{o}_j\}_{j=1}^{\tilde{T}}$, distance function $d(\cdot, \cdot)$, temperature term $\lambda$

`// Calc probability matrix`

$P_{i,j} = \exp(-\lambda d(o_t, \tilde{o}_j))$ for $t \in [[T]], j \in [[\tilde{T}]]$

`// Init coverage at learner time ` $t = 1$

$C_{1,1} = P_{1,1}$

$C_{1,j} = C_{1,j-1} P_{1,j}$ for $j \in \{2, 3, \ldots, \tilde{T}\}$

`// Init coverage of first demo frame ` $j = 1$

$C_{t,1} = \max\{C_{t-1,1}, P_{t,1}\}$ for $t \in \{2, 3, \ldots, T\}$

`// Solve recurrence relation (5)`

**for** $t = 1$ to $T$ **do**

   **for** $j = 1$ to $T'$ **do**

      $C_{t,j} = \max\{C_{t-1,j}, C_{t,j-1} P_{t,j}\}$

   **end for**

**end for**

`// Compute the final ORCA rewards (6)`

**Return** Rewards $\mathcal{R}_{\text{ORCA}} = \{C_{t,\tilde{T}-1} P_{t,\tilde{T}} \mid t \in [[T]]\}$

---

### 4.2. Analysis

The `ORCA` reward satisfies the two desiderata for a sequence-matching reward function (Sec. 3), overcoming the failure modes of frame-level matching algorithms. In the following analysis, we define a subgoal $\tilde{o}_j$ to be *occupied* if $P_{t,j} \approx 1$.

**Proposition 4.1 (`ORCA` enforces subgoal ordering).** *Let* $\xi^-$ *be a trajectory that is out of order; specifically, there exists a subgoal* $\tilde{o}_j$ *such that* $\tilde{o}_j$ *is occupied at time* $t$ *but* $\tilde{o}_{j-1}$ *is not yet occupied. Let* $\xi^+$ *be a trajectory that is identical to* $\xi^-$, *except that it occupies* $\tilde{o}_{j-1}$ *before time* $t$. *Then,* $\mathcal{R}_{ORCA}(o_t^+, \xi^+, \tilde{\xi}) > \mathcal{R}_{ORCA}(o_t^-, \xi^-, \tilde{\xi})$.

By (5), at a timestep, the ordered coverage of the current subgoal can be no greater than the coverage of the previous subgoal. Since $\xi^+$ occupies $\tilde{o}_{j-1}$ before $t$ and $\xi^-$ does not, $\xi^+$ achieves a greater coverage of the subgoal $\tilde{o}_{j-1}$ than $\xi^-$. The trajectories are otherwise equivalent, so $\xi^+$ must achieve a higher `ORCA` reward at time $t$. A formal proof is given in Appendix A.1. This overcomes OT's failure mode (Sec. 3.1).

**Proposition 4.2 (`ORCA` enforces subgoal coverage).** *Let* $\xi^-$ *be a trajectory that occupies* $\tilde{o}_{j-1}$ *at time* $t-1$ *and continues to occupy* $\tilde{o}_{j-1}$ *at time* $t$, *instead of progressing towards* $\tilde{o}_j$. *Let* $\xi^+$ *be an identical trajectory that progresses towards* $\tilde{o}_j$ *at* $t$, *and assume that neither trajectory has been closer to* $\tilde{o}_j$ *before. Then,* $\mathcal{R}_{ORCA}(o_t^+, \xi^+, \tilde{\xi}) > \mathcal{R}_{ORCA}(o_t^-, \xi^-, \tilde{\xi})$.

Both trajectories achieve the same coverage of subgoals up to $\tilde{o}_{j-1}$. Since $\xi^+$ moves closer to the next subgoal $\tilde{o}_j$ than $\xi^-$, it achieves a higher probability of occupying $\tilde{o}_j$ and gets higher coverage of $\tilde{o}_j$. The trajectories are otherwise equivalent, so $\xi^+$ must achieve a higher `ORCA` reward at

time $t$. A formal proof is given in Appendix A.1. This overcomes the failure modes of DTW and TemporalOT in Sec. 3.2 and 3.3. Appendix A.2 visualizes how the `ORCA` reward avoids the example failures of these frame-level matching algorithms.

### 4.3. Pretraining

We observe that in practice, `ORCA` can have multiple local minima. This is because the agent is trying to achieve high coverage for many subgoals. It has a trade-off between covering all subgoals equally well, which is often slower, or quickly achieving high coverage for most subgoals while a small portion of them get lower (but still nonzero) coverage. Some of these minima are undesirable, e.g. if the small portion of subgoals that the agent only partially covers are vital to the task and require the agents to match them more perfectly, the agent is more likely to fail.

To initialize the agent in a better basin, we first bias the agent towards spending an equal amount of time attempting to cover each subgoal, and then train with the `ORCA` reward. Specifically, we pretrain the agent on a reward function that assumes the video demonstration is temporally aligned (e.g., `TemporalOT` rewards). In Section 5, we empirically show the importance of pretraining.

## 5. Experiments

### 5.1. Experimental Setup

**Environments.** We evaluate our approach across two environments (details in Appendix B):

- **Meta-World (Yu et al., 2020).** Following Fu et al. (2024c), we use ten tasks from the Meta-world environment to evaluate the effectiveness of `ORCA` reward in the robotic manipulation domain. On each rollout, the locations of task-relevant objects are randomized according to the default setup of Metaworld. The training and evaluation seeds are also different. We classify the tasks into three difficulty levels based on the number of types of motions required and whether the task requires precision. Visual demonstrations are generated using hand-engineered policies provided by the environment.
- **Humanoid.** We define four tasks in the MuJoCo `Humanoid-v4` environment (Todorov et al., 2012) to examine how well `ORCA` works with precise motion. Because there is no predefined expert, we obtain visual demonstrations by rendering an interpolation between the initial and goal joint state.

**RL Policy.** For Meta-world, we follow the RL setup in Fu et al. (2024c). We train DrQ-v2 (Yarats et al., 2021) with state-based input for 1M steps and evaluate the policy every 10k steps on 10 randomly seeded environments. For

*Table 1.* **Meta-world results on temporally *misaligned* demonstrations.** We report the mean expert-normalized returns with standard error, and we highlight the top-performing approaches. Multiple are included if they are within the standard error of the top score. Agents trained with ORCA consistently outperform other frame-level matching approaches. RoboCLIP is omitted because it fails for all tasks.

| Category | Environment | Threshold | DTW | OT | TemporalOT | ORCA (NP) | **ORCA** |
|---|---|---|---|---|---|---|---|
| Easy | Button-press | 0.30 (0.10) | 0.00 (0.00) | 0.00 (0.00) | 0.10 (0.02) | 0.45 (0.11) | **0.62 (0.11)** |
| | Door-close | 0.34 (0.07) | 0.00 (0.00) | 0.00 (0.00) | 0.19 (0.01) | 0.86 (0.01) | **0.88 (0.01)** |
| Medium | Door-open | 0.00 (0.00) | 0.00 (0.00) | 0.00 (0.00) | 0.08 (0.01) | **1.60 (0.09)** | 0.89 (0.13) |
| | Window-open | 0.72 (0.14) | 0.00 (0.00) | 0.19 (0.06) | 0.26 (0.05) | **0.86 (0.17)** | **0.85 (0.16)** |
| | Lever-pull | 0.07 (0.02) | 0.00 (0.00) | 0.00 (0.00) | 0.07 (0.03) | **0.27 (0.08)** | **0.28 (0.09)** |
| | Hand-insert | 0.00 (0.00) | 0.00 (0.00) | 0.03 (0.02) | 0.00 (0.00) | **0.08 (0.08)** | **0.04 (0.04)** |
| | Push | 0.07 (0.05) | 0.00 (0.00) | 0.03 (0.01) | 0.01 (0.01) | 0.02 (0.02) | 0.00 (0.00) |
| Hard | Basketball | 0.00 (0.00) | 0.00 (0.00) | 0.00 (0.00) | 0.01 (0.01) | **0.07 (0.03)** | 0.01 (0.00) |
| | Stick-push | 0.12 (0.04) | 0.00 (0.00) | 0.07 (0.02) | 0.36 (0.00) | 0.46 (0.13) | **1.25 (0.04)** |
| | Door-lock | 0.00 (0.00) | 0.05 (0.02) | 0.04 (0.02) | 0.00 (0.00) | **0.23 (0.09)** | **0.19 (0.08)** |
| | **Average** | 0.16 (0.02) | 0.01 (0.00) | 0.04 (0.01) | 0.11 (0.01) | **0.49 (0.04)** | **0.50 (0.04)** |

*Table 2.* **Humanoid results on temporally misaligned demonstrations.** Results are presented as the mean returns with standard error. TemporalOT is abbreviated to TOT. For results of all baselines, see Table 4 in Appendix C.

| Task | TOT | ORCA (NP) | **ORCA** |
|---|---|---|---|
| Arm up (L) | 5.29 (2.22) | 65.9 (8.25) | **81.6 (3.65)** |
| Arm up (R) | 7.67 (2.88) | **92.5 (4.71)** | 49.6 (5.00) |
| Arms out | 1.62 (0.75) | **72.7 (10.1)** | 8.50 (2.60) |
| Arms down | 11.6 (3.56) | 19.7 (5.03) | **33.4 (7.20)** |
| Average | 6.55 (2.35) | **62.9 (7.02)** | 43.3 (4.61) |

the Humanoid environment, we train SAC (Haarnoja et al., 2018) for 2M steps and evaluate the policy every 20k steps on 8 environments. All policies use state-based input, and in metaworld we include an additional feature that represents the percentage of total timesteps passed. Appendix B.4 contains RL training details and hyperparameters.

**Baselines.** We compare ORCA against baselines that use frame-level matching algorithms: OT (Tian et al., 2024), TemporalOT (Fu et al., 2024c), and DTW (Sakoe & Chiba, 1978). We also compare ORCA, which is pretrained on TemporalOT rewards for half of the total timesteps and ORCA rewards for the remaining timesteps, against ORCA(NP), which fully trains on ORCA rewards without any initialization. All approaches use the pretrained ResNet50 (He et al., 2016) to extract visual features and cosine similarity as the distance function. We include a simple baseline Threshold, which tracks the subgoals completed based on a threshold distance, and a transformer-based approach RoboCLIP (Sontakke et al., 2024), which directly encodes an entire video. Details are in Appendix B.5. We also investigate a language-conditioned baseline and a traditional IRL

baseline on a subset of the tasks in Appendix C.3.

**Metrics.** We evaluate the final checkpoints of all approaches on cumulative binary rewards, or **returns**, so a policy that succeeds quickly and remains successful is better. In Metaworld, we use the ground-truth sparse rewards and report the normalized return, which is the return as a fraction of the expert's return on the same task, given the same number of timesteps. We define a success metric for Humanoid, using privileged states, which no approaches have access to. An agent is successful if it can remain standing (torso height above 1.1) and its arm joint position is close to the goal joint position (Euclidean distance less than 1). There is no expert in for Humanoid tasks, so we report unnormalized returns.

### 5.2. How well does ORCA perform given temporally misaligned demonstrations?

**Metaworld.** For each original demonstration, we subsample it such that the first one-fifth retains the original execution speed, but the rest is sped up by five to ten times. In Table 4.3, the ORCA reward significantly outperforms other baselines and trains agents to acquire the highest average normalized return of 0.50. The *Push* task is difficult for all approaches and the only one where ORCA or ORCA(NP) does not achieve top performance. As analyzed by Fu et al. (2024d), the distance matrix is generally noisier for the *Push* task due to the small size of the target object, which benefits a sparser reward function like Threshold. Meanwhile, rewards based on frame-level matching algorithms perform poorly on all tasks, revealing their weakness when encountering temporal misalignment.

**Humanoid.** Because there does not exist an expert policy for the Humanoid tasks, we generate demonstrations by interpolating ten frames between the initial and final joint

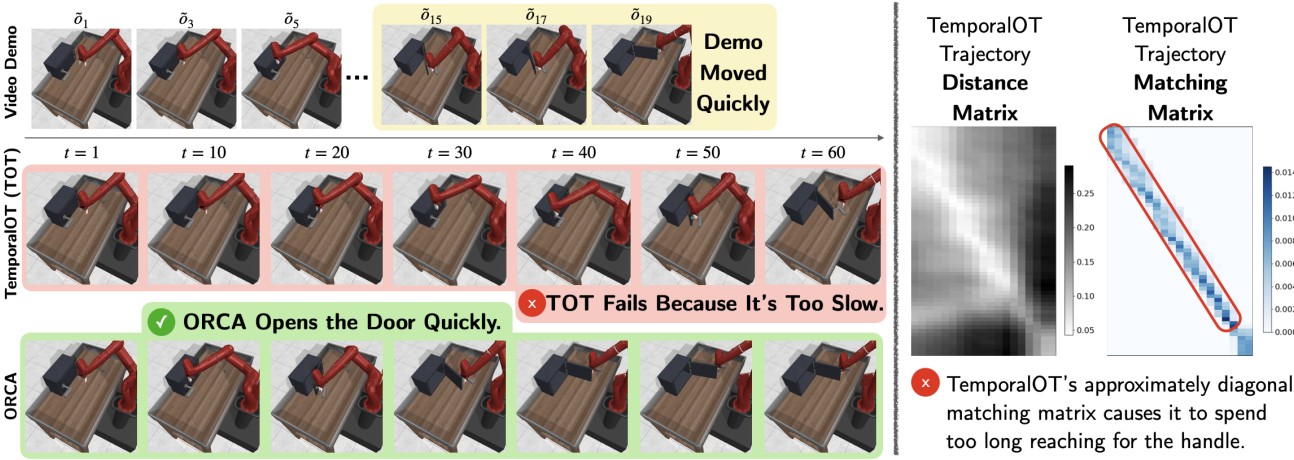

*Figure 3.* **Qualitative example of `TemporalOT` failing to encourage full subgoal coverage.** The video demonstration shows how to open a door by latching on the door handle, but it speeds through the movement after latching. `TemporalOT` trains a slow agent that fails to complete the task due to its diagonal-like matching matrix, but `ORCA` trains a successful agent that completes the task efficiently.

positions. These demonstrations are naturally temporally misaligned because the environment is unstable, making it impossible to follow the subgoals at the same speed. Table 2 shows that agents trained with `ORCA(NP)` achieve the highest average cumulative return of 62.9. Due to poor `TemporalOT` performance, `ORCA` does not benefit from pretraining, and `ORCA(NP)` thus performs better because it is trained on the ordered coverage reward for more steps. We further investigate the failure of `TemporalOT` in Sec. 5.4. Meanwhile, Fig. 15 in Appendix C shows how the `ORCA` reward satisfies Prop. 4.1 and 4.2, covering the subgoals as quickly as possible and successfully completing the task.

**Varying Misalignment Level.** We identify two types of temporal misalignment: either a slower demonstration that contains pauses, or a faster demonstration that accelerates through a segment. For each misalignment type, we randomly perturb the original demonstrations of three Metaworld tasks (*Door-open*, *Window-open*, *Lever-pull*), where the misalignment level controls how varied and nonlinear speed changes are. See appendix B.6 for details.

In Fig. 4, `ORCA` consistently maintains a higher return compared to `TemporalOT` as the demonstrations become more misaligned. Meanwhile, `TemporalOT`'s performance significantly deteriorates when there is any level of misalignment. When the demonstrations are sped up, because `TemporalOT` encourages agents to spend an equal amount of time at each subgoal (in-depth discussion in Sec 5.4), `TemporalOT` agents often cannot finish the task in time. This problem is further exacerbated when the demonstrations are slowed down and longer than the learner trajectory. `ORCA`'s performance also worsens more given slower demonstrations compared to faster ones.

In addition to being affected by poor initialization due to

`TemporalOT`'s poor performance, we observe that when `ORCA` agents fail, they successfully follow the general motions of the demonstration, but they miss details (e.g., aligning the gripper with the target object). We hypothesize that this behavior is caused by the frame-level distance metric, which pays more attention to the general robot arm motions than details, allowing most subgoals to achieve relatively good coverage.

### 5.3. How important is enforcing subgoal ordering?

Fig. 17 in Appendix C.1 shows a key failure point for `OT`: its matching matrix can match later subgoals to earlier learner frames and vice versa. The optimal matching matrix minimizes the transport cost regardless of the order in which subgoals are completed, thereby giving higher `OT` rewards to trajectories that violate the temporal ordering. In both Metaworld and Humanoid environments, the OT rewards create a difficult optimization landscape that causes the agents to learn undesirable behaviors. Fig. 17 also demonstrates that `TemporalOT` can violate temporal ordering depending on the demonstration length and the mask window size. Tuning this value is a major drawback of `TemporalOT` because it requires prior knowledge of the temporal alignment.

### 5.4. How important is enforcing full subgoal coverage?

Fig. 3 shows that `ORCA` successfully trains an efficient agent, while the `TemporalOT` agent opens the door much more slowly and fails. The `OT`/`TemporalOT` formulation assumes that the learner and demonstration distributions are uniform in time. However, given a temporally misaligned demonstration, different portions of learner frames should be matched to different subgoals, which is impossible using this formulation. Empirically, successful trajectories

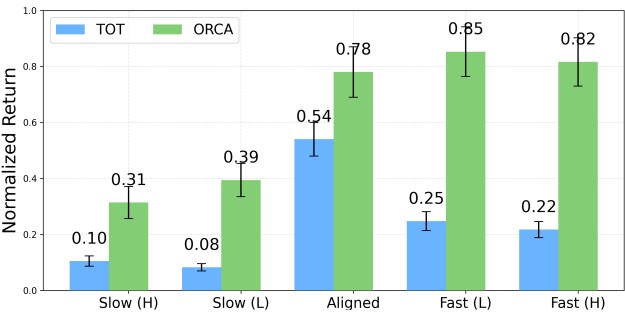

*Figure 4.* **Results given varying levels of temporal misalignment.** We report the mean expert-normalized returns with standard error across 3 Meta-world tasks (Door open, Window open, Lever pull). We generate 3 perturbed demonstrations per task per misalignment level (L=Low, H=High), training on each demonstration separately.

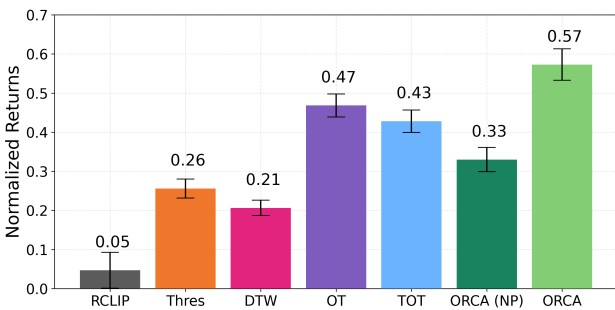

*Figure 5.* **Results given temporally *aligned* demonstrations**. We report mean expert-normalized returns with standard error across all 10 Meta-world tasks.

produce coupling matrices that approximate the diagonal matrix, as shown in Fig. 3. The subsequent rewards teach the agent to spend an equal amount of time at each subgoal instead of following the demonstration as fast as possible, which can cause the agent to exhaust the timesteps before completing the task. `TemporalOT` also exhibits poor performance for Humanoid tasks. We hypothesize that the agent fails because `TemporalOT` rewards force the agent to spend an equal amount of time in each intermediate subgoal, which is difficult in a highly unstable environment. We show `DTW`'s failure case in Fig. 18 of Appendix C.1.

### 5.5. How does pretraining affect ORCA's performance?

Pretraining leads to better `ORCA` performance when the pretraining strategy is able to obtain some success. In Table. 4.3, `ORCA` is equal or better than `ORCA(NP)` on temporally misaligned demonstrations. In Humanoid tasks, `ORCA(NP)` achieves higher overall performance because `TemporalOT` almost entirely fails on every task. The effect of pretraining is most apparent when the demonstrations are temporally aligned because this setting satisfies the core assumption of `TemporalOT`. In Fig. 5, `ORCA` achieves an

average normalized return of $0.57$ compared to `ORCA(NP)` $(0.33)$ on aligned demonstrations.

`ORCA(NP)` fails due to undesirable local minima. Consider the *Stick-Push* task in Fig. 19 of Appendix C.1, where the robot arm needs to grasp the stick before pushing the water bottle with that stick. The `ORCA(NP)` policy directly moves to push the water bottle without the stick. Because the robot arm initially seems close to the stick, the `ORCA(NP)` policy could get partial coverage on earlier subgoals while collecting high coverage on later subgoals for pushing the bottle. Meanwhile, `ORCA` is initialized with the `TemporalOT` policy that fails the task but spends equal time attempting each subgoal. `ORCA` is able to refine the policy, finding the middle ground between `ORCA(NP)` and `TemporalOT` and quickly grasping the stick before pushing the water bottle with it. Overall, pretraining initializes `ORCA` in a better basin, allowing it to train successful and efficient policies.

### 5.6. Does ORCA scale with more demonstrations?

Following the strategy of Fu et al. (2024c), `ORCA` can be adapted to multiple demonstration videos $\tilde{\Xi} = \{\tilde{\xi}^1, \ldots, \tilde{\xi}^N\}$ by calculating the `ORCA` reward at each timestep with respect to every video and max-pooling:

$$\mathcal{R}_{\text{ORCA}}(o_t, \xi, \tilde{\Xi}) = \max_{\tilde{\xi} \in \tilde{\Xi}} \mathcal{R}_{\text{ORCA}}(o_t, \xi, \tilde{\xi}) \qquad (7)$$

To study how the number of demonstrations affects performance, we first investigate the setting where they all have the same speed, and the demonstrations only vary due to randomly initialized object locations. Figure 6 shows how `ORCA`'s performance improves as the number of demonstrations increases. In contrast, although `TemporalOT` handles multiple videos using the same method, and it initially benefits from having more than one demonstration, its performance starts to degrade given a large number of demonstrations.

We additionally train policies using 4 demonstrations with different speeds by randomly sampling one video from each of the four temporal misalignment categories described in Figure 4. `ORCA` outperforms `TemporalOT` in this scenario, as shown in figure 6. `ORCA` policies additionally perform better when trained on 4 demonstrations with different speeds than just a single demonstration $(0.67 \pm 0.08 \rightarrow 0.77 \pm 0.08)$, although they are still worse than 4 same speed demonstrations. Meanwhile, `TemporalOT` shows minimal gains with different speed demonstrations $(0.14\pm0.02 \rightarrow 0.16\pm0.02)$, demonstrating its inability to handle temporal misalignment at scale. Overall, `ORCA` consistently benefits from training with multiple demonstrations and exhibits greater robustness compared to baselines when these demonstrations have different speeds.

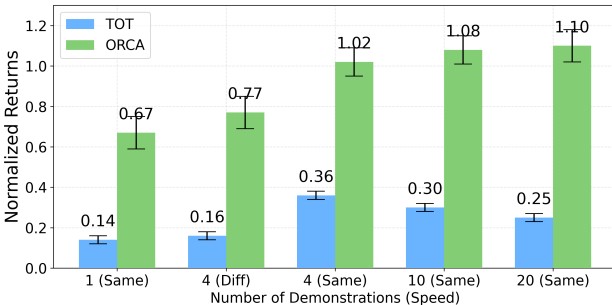

*Figure 6.* **Results given more demonstrations**. *Same* indicates that all demonstrations in the set are the same speed, whereas *Diff* indicates that they are different speeds. We report mean expert-normalized returns with standard error across 3 Meta-world tasks (Door open, Window open, Lever pull).

### 5.7. How does ORCA perform given a policy conditioned on image observations?

ORCA's goal is to estimate rewards from a single demonstration in the observation space before learning a policy, making no assumptions about the policy itself. We additionally evaluate ORCA when the policy is conditioned on image observations instead of ground-truth states, following the DrQv2 setup of (Fu et al., 2024c). The results in Appendix C.5 demonstrate that the state-based and image-based policies trained with ORCA have similar performance (average normalized return: $0.704 \pm 0.10 \rightarrow 0.76 \pm 0.06$). In contrast, policies trained with TemporalOT perform poorly, regardless of their input.

### 5.8. Visual Encoder Ablation

ORCA works with any visual encoder that can act as a distance metric between frames. We include in Appendix C.4 an additional ablation of ORCA with LIV (Ma et al., 2023), a robotics-specific visual encoder, and DINOv2 (Oquab et al., 2023), a standard vision model. The Resnet50 used for the main result in this paper achieves the best performance. However, there is high variability, substantiating the findings of prior work that evaluates different visual encoders for RL training (Hu et al., 2023). Ultimately, the best visual encoder is a practical and task-dependent choice.

## 6. Related Works

**Learning From a Few Video Demonstrations.** Prior works in imitation learning (Duan et al., 2017; Fu et al., 2024a; Jain et al., 2024; Palo & Johns, 2024; Xu et al., 2022) train a policy that, at inference time, takes a video-action demonstration and the current robot state to output actions. This is different from our policy formulation because ORCA and its baselines do not have access to demonstrations with action labels. Closer to our setting, some works focus on training

a reward model that takes videos as input and outputs scalar rewards for RL (Sontakke et al., 2024; Yang et al., 2024), but these models often require fine-tuning on task-specific data to improve performance (Fu et al., 2024b). We follow IRL's formulation where we aim to match the learner and demonstration distributions, which is equivalent to optimizing the Integral Probability Metrics (IPMs) (Sun et al., 2019; Swamy et al., 2021).

**Optimal Transport Used In IRL.** Recent works leverage OT (Peyré & Cuturi, 2020) to optimize IPMs. In the imitation learning setting, where a teleoperated demonstration dataset contains state-action labels, prior works can directly minimize the Wasserstein distance between the learner and the demonstration's state-action distributions (Bobrin et al., 2024; Dadashi et al., 2021; Luo et al., 2023; Papagiannis & Li, 2022; Xiao et al., 2019). These approaches do not work given only a single visual demonstration. Without access to privileged states, recent works instead use a distance function that measures the transport cost between two visual embeddings (Cohen et al., 2022; Fu et al., 2024c; Guzey et al., 2024; Haldar et al., 2023a;b; Kedia et al., 2025; Liu et al., 2024; Tian et al., 2024). Both types of approaches assume that a Markovian function measuring transport cost between two timesteps is sufficient. However, our work tackles sequence-matching problems, where agents must follow the subgoals from a temporally misaligned demonstration in the correct order. OT with a Markovian distance function fails because the true reward function depends on the entire trajectory, a limitation that our approach addresses.

## 7. Discussion

We investigate sequence-matching tasks, where the learner must follow a expert video demonstration that may be temporally misaligned. We analyze how algorithms that match the learner and expert distribution at the frame level (e.g., optimal transport) result in reward functions that fail in this setting. Following our key insight that matching should be defined at the sequence level, we present ORCA: a principled reward function that computes the probability that the learner has covered every subgoal in the correct order. Experiments on Meta-world and Humanoid tasks show that ORCA rewards train agents that complete the tasks efficiently regardless of the level of temporal misalignment.

We recognize a few limitations of ORCA: (1) it relies on a good visual distance metric that measures the similarity between two frames. Future work will explore using online finetuning to improve the encoder (Fu et al., 2024b) and solve cross-embodiment tasks where temporal misalignment is common. (2) video-following may result in unexpected failures due to task misspecification. We plan to explore how ORCA can be applied to language subgoals in addition to video frames.

## Acknowledgments

This work was supported in part by the National Science Foundation FRR (#2327973).

## Impact Statement

This paper presents work whose goal is to advance the field of Machine Learning. There are many potential societal consequences of our work, none which we feel must be specifically highlighted here.

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

# Appendix

## Table of Contents

## A. Detailed Analysis of ORCA

We prove propositions 4.1 and 4.2, showing that the ORCA reward encourages the agent to complete all subgoals in the correct order, thus addressing the limitations of baselines that use frame-level matching. First, we derive basic properties of the ORCA coverage matrix. Then, we restate the propositions and prove them using these properties.

### A.1. Proofs of ORCA Desiderata

**Lemma A.1.** *For all t, $C_{t,j} = \max_{i=1}^{t} C_{i,j-1} P_{i,j}$.*

*Proof.* We prove this by induction on $t$.

**Base case:** For $t = 1$, $C_{1,j} = C_{1,j-1} P_{1,j}$ by (5), which satisfies the statement.

**Inductive step:** Assume $C_{t,j} = \max_{i=1}^{t} C_{i,j-1} P_{i,j}$ holds for $t$. For $t + 1$, by the recursive definition (5):

$$C_{t+1,j} = \max\{C_{t,j}, C_{t+1,j-1} P_{t+1,j}\}.$$

Substituting the inductive hypothesis:

$$C_{t+1,j} = \max\{\max_{i=1}^{t} C_{i,j-1} P_{i,j}, C_{t+1,j-1} P_{t+1,j}\} = \max_{i=1}^{t+1} C_{i,j-1} P_{i,j}.$$

Thus, the statement holds for $t + 1$. By induction, the lemma is proven.  □

**Corollary A.2.** *If at time t, $\max_{i=1}^{t} P_{i,j} = P_{t,j}$, then $C_{t,j} = C_{t,j-1} P_{t,j}$.*

*Proof.* By the non-decreasing property of coverage along the learner axis (5),

$$\max_{i=1}^{t} C_{i,j-1} = C_{t,j-1}. \tag{8}$$

By lemma A.1,

$$C_{t,j} = C_{t,j-1} P_{t,j}. \tag{9}$$

$\square$

**Corollary A.3.** *If two trajectories $\xi^+$ and $\xi^-$ are identical, except a subgoal $\tilde{o}_j$ is covered at time $t$ in $\xi^+$ and is not covered in $\xi^-$, then $C_{t,j}^+ > C_{t,j}^-$.*

*Proof.* $\xi^+$ achieves coverage of $\tilde{o}_j$ at time $t$, so by corollary A.2:

$$C_{t,j}^+ = C_{t,j-1}^+ P_{t,j}^+ > C_{t-1,j}^+. \tag{10}$$

Since the trajectories are identical prior to $t$, we have:

$$C_{t-1,j}^+ = C_{t,j-1}^-. \tag{11}$$

Moreover, $P_{t,j}^+ > P_{t,j}^-$ implies

$$C_{t,j-1}^+ P_{t,j}^+ > C_{t,j-1}^- P_{t,j}^-. \tag{12}$$

It follows that

$$C_{t,j-1}^+ P_{t,j}^+ > \max\{C_{t,j-1}^-, C_{t,j-1}^- P_{t,j}^-\}. \tag{13}$$

We conclude from the coverage definition (5):

$$C_{t,j}^+ > C_{t,j}^-. \tag{14}$$

$\square$

**Proposition A.4** (**ORCA enforces subgoal ordering** (Restating Prop. 4.1))**.** *Let $\xi^-$ be a trajectory that is out of order; specifically, there exists a subgoal $\tilde{o}_j$ such that $\tilde{o}_j$ is occupied at time $t$ and $\tilde{o}_{j-1}$ is not yet covered. Let $\xi^+$ be a trajectory that is identical to $\xi^-$, except that it covers $\tilde{o}_{j-1}$ before time $t$. Then, $\mathcal{R}_{ORCA}(o_t^+, \xi^+, \tilde{\xi}) > \mathcal{R}_{ORCA}(o_t^-, \xi^-, \tilde{\xi})$.*

*Proof.* Because $\tilde{o}_j$ is occupied at time $t$ in $\xi^+$ ($\max_{i=1}^{t} P_{i,j}^+ = P_{t,j}^+$), by corollary A.2,

$$C_{t,j}^+ = C_{t,j-1}^+ P_{t,j}^+. \tag{15}$$

According to the DP recurrence relation (5), there are two cases for $C_{t,j}^-$:

**Case 1.** $C_{t,j}^- = C_{t-1,j}^-$

By lemma A.1,

$$C_{t-1,j}^- = \max_{i=1}^{t-1} C_{i,j-1}^- P_{i,j}^-. \tag{16}$$

By corollary A.3, and the fact that coverage is nondecreasing along the demonstration,

$$C_{t,j-1}^+ > C_{t,j-1}^- \geq \max_{i=1}^{t} C_{i,j-1}^-. \tag{17}$$

Because $\xi^+$ and $\xi^-$ both occupy subgoal $\tilde{o}_j$ at time $t$:

$$P_{t,j}^+ = \max_{i=1}^{t} P_{i,j}^-. \tag{18}$$

Multiplying (17) and (18) lets us establish a bound on (16):

$$C_{t,j-1}^+ P_{t,j}^+ > \max_{i=1}^{t} C_{i,j-1}^- \max_{i=1}^{t} P_{i,j}^- \geq \max_{i=1}^{t-1} C_{i,j-1}^- P_{i,j}^-. \tag{19}$$

Substituting (15), we get:

$$C_{t,j}^+ > C_{t,j}^-. \tag{20}$$

**Case 2.** $C_{t,j}^- = C_{t,j-1}^- P_{t,j}^-$

Because $P_{t,j}^+ = P_{t,j}^-$, and by corollary A.3,

$$C_{t,j-1}^+ P_{t,j}^+ > C_{t,j-1}^- P_{t,j}^-. \tag{21}$$

Substituting (15), we get:

$$C_{t,j}^+ > C_{t,j}^-. \tag{22}$$

Since $C_{t,j}^+ > C_{t,j}^-$ in both cases, and the trajectories are otherwise identical,

$$\mathcal{R}_{ORCA}(o_t^+, \xi^+, \tilde{\xi}) > \mathcal{R}_{ORCA}(o_t^-, \xi^-, \tilde{\xi}). \tag{23}$$

$\square$

**Proposition A.5** (**ORCA** enforces subgoal coverage. (Restating Prop. 4.2)). *Let $\xi^-$ be a trajectory that occupies $\tilde{o}_{j-1}$ at time $t-1$ and continues to occupy $\tilde{o}_{j-1}$ at time $t$, instead of progressing towards $\tilde{o}_j$. Let $\xi^+$ be an identical trajectory that progresses towards $\tilde{o}_j$ at $t$, and assume that neither trajectory has been closer to $\tilde{o}_j$ before. Then, $\mathcal{R}_{ORCA}(o_t^+, \xi^+, \tilde{\xi}) > \mathcal{R}_{ORCA}(o_t^-, \xi^-, \tilde{\xi})$.*

*Proof.* At time $t$, $\xi^+$ moves closer to $\tilde{o}_j$ than it has previously been. Thus, by corollary A.2:

$$C_{t,j}^+ = C_{t,j-1}^+ P_{t,j}^+ > C_{t-1,j}^+. \tag{24}$$

There are two cases for $C_{t,j}^-$.

**Case 1.** $C_{t,j}^- = C_{t-1,j}^-$

Because $\xi^+$ is identical to $\xi^-$ prior to $t$,

$$C_{t-1,j}^+ = C_{t-1,j}^- \tag{25}$$

Thus, by (24),

$$C_{t,j}^+ > C_{t-1,j}^+ = C_{t-1,j}^- = C_{t,j}^-. \tag{26}$$

**Case 2.** $C_{t,j}^- = C_{t,j-1}^- P_{t,j}^-$

Since $\xi^+$ is identical to $\xi^-$ prior to $t$, and at time $t$ neither improves its coverage of subgoals prior to $\tilde{o}_j$:

$$C_{t,j-1}^+ = C_{t,j-1}^-. \tag{27}$$

However, because $\xi^+$ moves closer to $\tilde{o}_j$ than $\xi^-$ at time t,

$$P_{t,j}^+ > P_{t,j}^-. \tag{28}$$

We conclude that

$$C_{t,j}^+ = C_{t,j-1}^+ P_{t,j}^+ > C_{t,j-1}^- P_{t,j}^- = C_{t,j}^-. \tag{29}$$

Since $C_{t,j}^+ > C_{t,j}^-$ in both cases, and the trajectories are otherwise identical,

$$\mathcal{R}_{ORCA}(o_t^+, \xi^+, \tilde{\xi}) > \mathcal{R}_{ORCA}(o_t^-, \xi^-, \tilde{\xi}). \tag{30}$$

$\square$

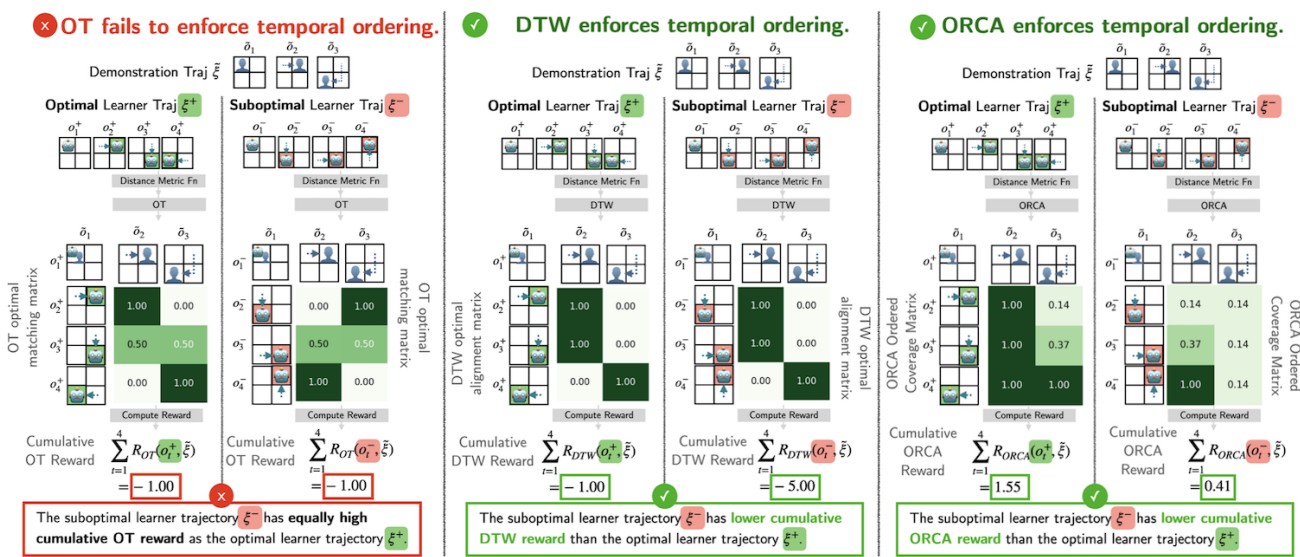

*Figure 7.* **Failure cases for OT reward in the 2D-Navigation environment. Both DTW and `ORCA` overcomes OT's limitation.**

### A.2. Toy Examples of `ORCA` Overcoming Failure Cases of Existing Approaches

We present complete figures showing how `ORCA` overcomes OT's failure to enforce subgoal ordering and DTW/Temporal-OT's failure to enforce full subgoal coverage. The distance function between each state is the Manhattan distance, which is Markovian.

In Fig 7, the suboptimal learner trajectory completes the demonstration subgoals in the wrong order compared to the optimal learner trajectory that completes the task in the correct order. Because OT treats the learner and trajectory distribution as two unordered sets, and the distance function does not encode any temporal information, it fails to penalize the suboptimal trajectory, giving both equally high rewards. In contrast, because DTW enforces temporal constraint in its alignment, the final alignment is the same for both trajectories, resulting in a lower DTW reward for the suboptimal trajectory. Similarly, `ORCA` measures the probability that *all subgoals are covered in the correct order*. Although the suboptimal learner trajectory perfectly occupies the third subgoal at time 3, because it has not occupied the second subgoal, its overall ordered coverage is still low, thereby penalizing the suboptimal learner trajectory for covering out of order.

In Fig. 8, the suboptimal learner trajectory stalls at the second subgoal while the optimal trajectory makes consistent progress towards covering all subgoals. DTW fails because it does not constrain the number of learner frames that can be matched with each subgoal. Consequently, its alignment matrix matches most learner frames to the second subgoal, which has no cost since they perfectly occupy it, resulting in equally high DTW rewards for both trajectories. In contrast, because `ORCA` rewards depends on *all* subgoals to be covered, the suboptimal trajectory receives low rewards for most timestep since it has not covered the final subgoal.

In Fig. 9, the video demonstration is temporally misaligned because the agent can move one cell at a time, and the subgoals would require the agent to take multiple timesteps to reach. The suboptimal learner trajectory is slow, spending 2 timesteps at earlier subgoals and failing to solve the tasks in time. Meanwhile, the optimal learner trajectory makes consistent progress and succeeds. Because TemporalOT assumes that the learner and demonstration demonstrations are temporally aligned, the mask window that it then defines causes the coupling matrix to also approximate a diagonal matrix. Such coupling matrix would encourage the agent to spend equal amount of time matching each subgoal, thereby rewarding the suboptimal trajectory that does so perfectly even though they did not finish the task. In contrast, `ORCA`'s rewards depend on *all* subgoals to be covered, so the suboptimal trajectory receives low rewards for all timesteps since it has not covered the final subgoal.

## B. Environment Details

We run experiments in two different environments: Meta-world (Yu et al., 2020), a manipulation environment, and `Humanoid-v4` (Todorov et al., 2012), a more difficult control environment. Following prior work (Fu et al., 2024c), we

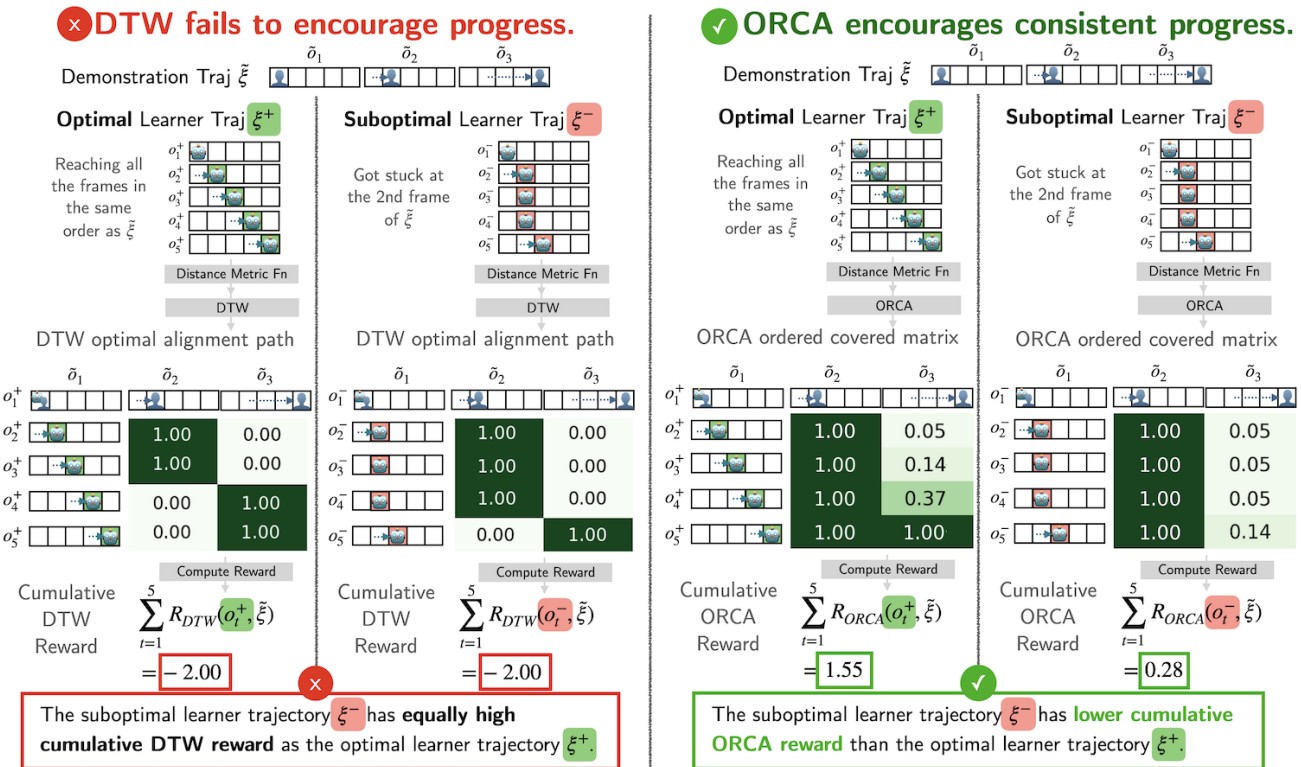

Figure 8. **Failure cases for DTW reward in the 2D-Navigation environment. ORCA overcomes DTW's limitation.** The suboptimal learner trajectory $\xi^-$ gets stuck at the second frame of the demonstration, while the optimal one $\xi^+$ makes consistent progress.

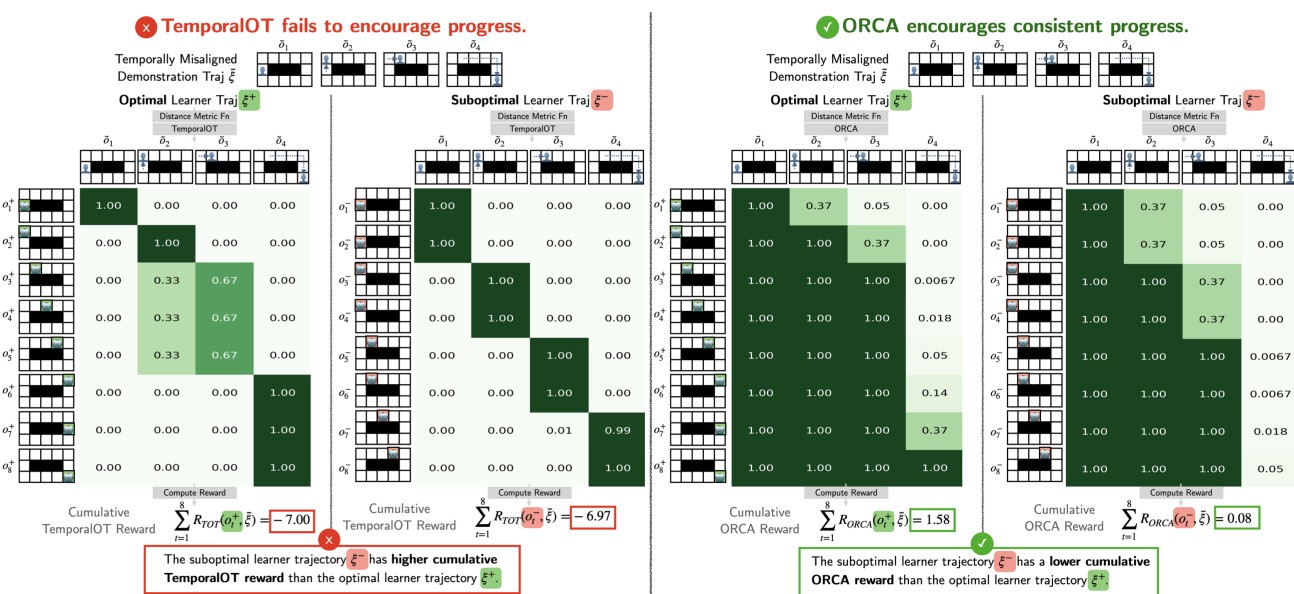

Figure 9. **Failure cases for TemporalOT reward given a temporally misaligned demonstration in the 2D-Navigation environment. ORCA overcomes TemporalOT's limitation.** the video demonstration is *temporally misaligned* because the agent can move one cell at a time, and the subgoals would require the agent to take multiple timesteps to reach. The suboptimal learner trajectory $\xi^-$ that moves slowly and fails to complete the task. In contrast, the optimal learner trajectory $\xi^+$ makes steady progress and completes the task successfully.

test 10 different tasks in Meta-world. To show that `ORCA` works in more general domains, we additionally design 4 tasks in the humanoid environment that require moving the arms of the humanoid.

### B.1. Visual Encoder

`ORCA`, as well as all baselines except `RoboCLIP`, requires a distance function defined on the space of images. We use a visual encoder to obtain embeddings for each image in the learner and expert trajectory, and find the pairwise distances between them to obtain the distance matrix.

In Meta-world, we follow prior work (Fu et al., 2024c) in using an off-the-shelf Resnet50 (He et al., 2016) as the visual encoder. We use the cosine distance between embeddings to produce a cost matrix.

In Humanoid, we find that off-the-shelf visual encoders do not capture the fine-grained details necessary for the more difficult control task. Instead, we train a model to predict the joint positions of the humanoid, and use the Euclidean distance between joint predictions as the distance function. To address out-of-distribution samples, we train a separate network to predict the model confidence, which is used to scale the final rewards. For more details on the joint predictor, see B.3.2.

### B.2. Meta-world

#### B.2.1. TASKS

We run experiments on 10 different tasks in the `Meta-world` (Yu et al., 2020) environment. In addition to the 9 tasks in Fu et al. (2024c), we added *Door-close*. We classify them into easy, medium, and hard based on factors like their visual difficulty, necessity for precise motor control, and interaction with other objects. For further information on the tasks, we refer the reader to (Fu et al., 2024c). Below is a brief description of the objective for each task:

| Task | Difficulty | Success Criteria |
|------|-----------|------------------|
| Button-press | Easy | The button is pushed. |
| Door-close | Easy | The door is fully closed. |
| Door-open | Medium | The door is fully open. |
| Window-open | Medium | The window is slid fully open. |
| Lever-pull | Medium | The lever is pulled up. |
| Hand-insert | Medium | The brown block is inserted into the hole in the table. |
| Push | Medium | The cylinder is moved to the target location. |
| Basketball | Hard | The basketball is in the hoop. |
| Stick-push | Hard | The bottle is pushed to the target location using the stick. |
| Door-lock | Hard | The locking mechanism is engaged (pushed down). |

#### B.2.2. TRAINING DETAILS

In order to improve performance across all methods, we employ two training strategies on top of the reward model and RL algorithm:

1) **Context embedding**: We use the context embedding-based cost matrix proposed in (Fu et al., 2024c), which can be interpreted as a diagonal smoothing kernel. Specifically, the distance between two frames is expressed as the average distance over the next $c_w$ learner and demonstration frames (where $c_w$ refers to the context window):

$$d_{window}(o_i^L, o_j^D) = \frac{1}{c_w} \sum_{k=1}^{c_w} d(o_{i+k}^L, o_{j+k}^D)$$

We choose a context window of length 3, which resulted in the best performance in (Fu et al., 2024c). Although a longer context window could damage performance on extremely mismatched tasks, we find that a small window helps regularize the noisiness of the visual distance metric.

2) **Timestep in agent state**: By nature of the sequence-following task, the reward at a given time step depends on the states visited by the learner in previous time steps. Thus, if the policy or value estimator cannot observe the entire trajectory, then

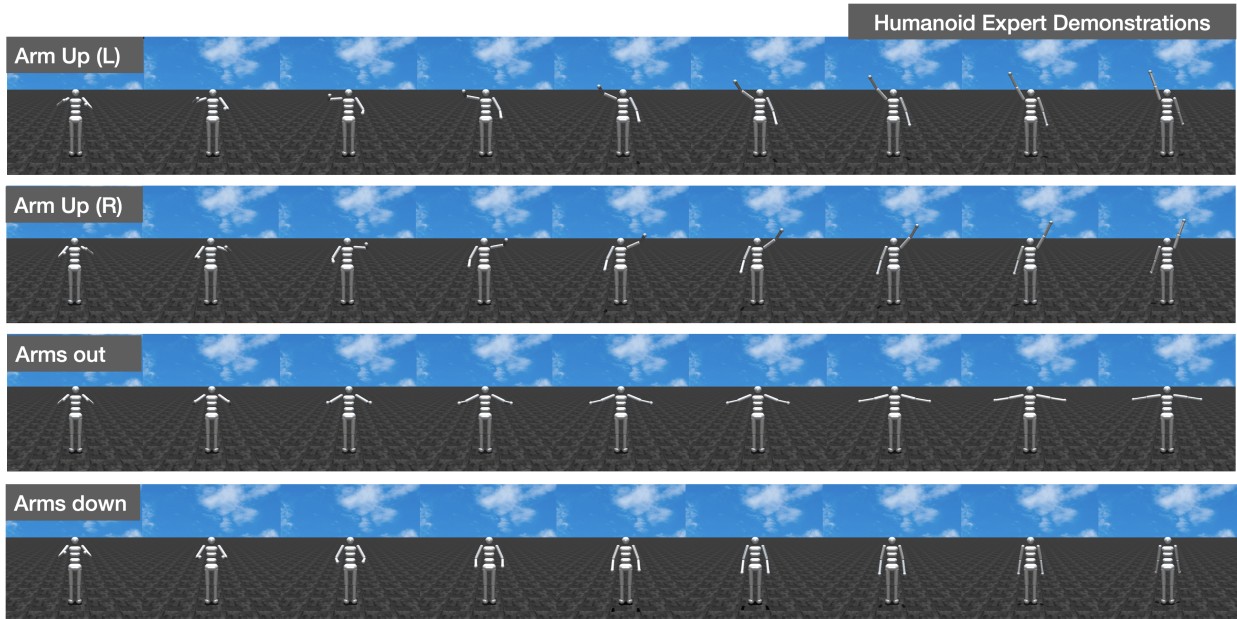

*Figure 10.* **Visual Demonstrations of the tasks in the Humanoid Environment.** These were generated by selecting the target final joint state, interpolating from the start joint state, and rendering the intermediate frames.

it does not have enough information to model the reward. In practice, we find that including the current time step (as a percentage of the episode length) in the state observation of the agent allows it to reasonably estimate the value function. We emphasize that, although the agent has access to the ground truth state and time step, *the reward model only sees the visual learner rollout and demonstration.* Because the purpose of this paper is to examine specific sequence-matching reward functions, we choose to use this empirical trick for our experiments, leaving further investigations into RL algorithms given non-Markovian rewards as future work.

### B.3. Humanoid

#### B.3.1. TASKS

We use the MuJoCo `Humanoid-v4` environment (Foundation, 2024; Todorov et al., 2012). In order to improve visual encoder performance, we modify the environment textures and camera angle, as described in (Rocamonde et al., 2024). At the beginning of an episode, the humanoid is spawned upright, slightly above the ground, with its arms curled towards its chest.

The humanoid's goal is to follow the motion of a demonstration trajectory within a maximum of 120 timesteps. We define 4 motions, corresponding to 4 demonstration trajectories. The humanoid must remain standing while doing all tasks. The visual demonstrations for each task are shown in B.3.1. These demonstration trajectories each have a length of 10 and are

| Task | Success Criteria |
|------|------------------|
| Arm up (L) | The left arm is raised above the head and the right arm is down. |
| Arm up (R) | The right arm is raised above the head and the left arm is down. |
| Arms out | Both arms are raised to shoulder height. (T-pose) |
| Arms down | Both arms are lowered to the side. |

generated by interpolating between the initial and final poses. Fig. 10 shows snapshots of these trajectories.

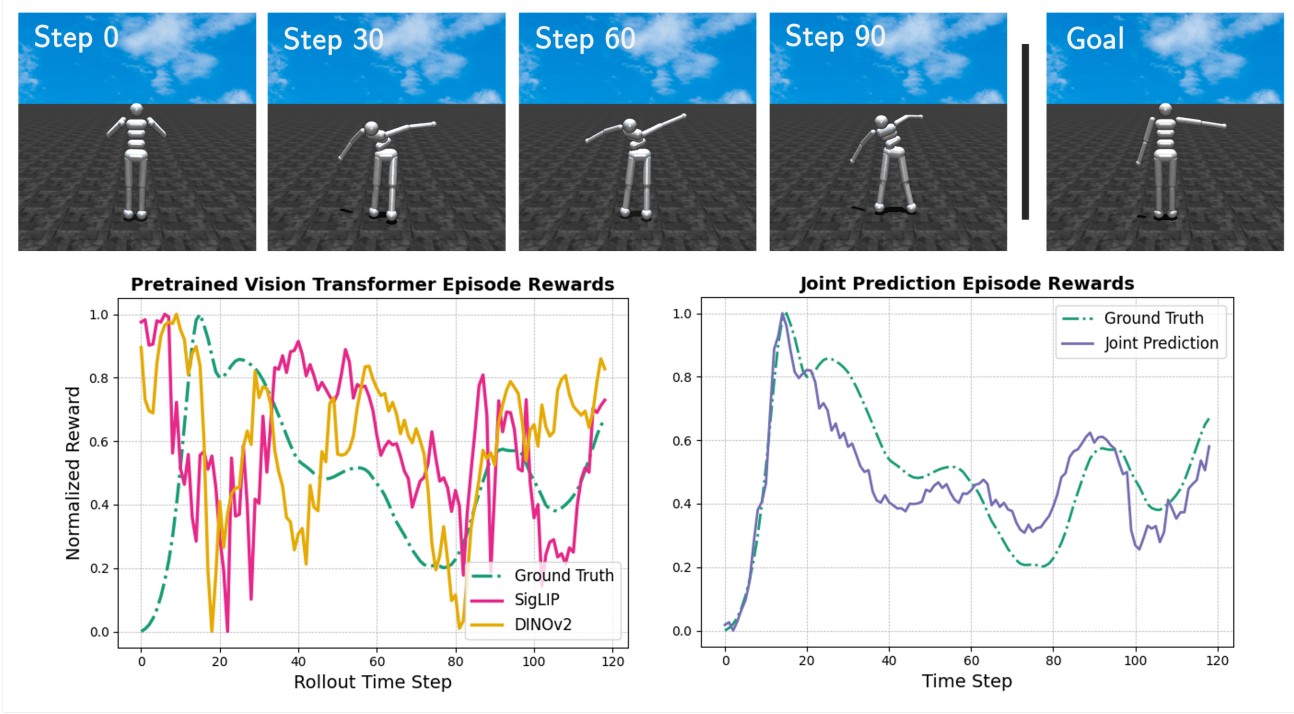

*Figure 11.* **Goal-reaching rewards of (left) two pretrained models and (right) our joint prediction model on an example learner trajectory, with the goal of raising the left arm to the side**. To emphasize their shape, all rewards are normalized along the trajectory dimension. Rewards using the pretrained SigLIP-ViT-B-16 (Zhai et al., 2023) and DINOv2-ViT-B-14-reg (Oquab et al., 2023) models are calculated as the cosine similarity between the learner and demonstration embeddings. The fine-tuned joint prediction model provides a smoother reward curve. The trajectories are in the MuJoCo `Humanoid-v4` environment (Foundation, 2024; Todorov et al., 2012), which is visually modified to mimic the setup of (Rocamonde et al., 2024).

### B.3.2. CONFIDENCE-SCALED VISUAL REWARDS.

Empirically, we found that off-the-shelf visual encoders produced noisy rewards in the Mujoco environment, as shown in Fig. 11. This resulted in training failure regardless of the distribution-matching or sequence-matching function. To solve this problem, we train a network that predicts the joint positions of the humanoid given an image observation. To address distribution shift during RL training, we use an autoencoder to predict a model confidence score, which we use to scale the final rewards. We additionally assume access to a stability reward function, which includes a control cost and a reward for remaining standing:

$$\mathcal{R}_{stability} = \exp(-(h_{torso} - 1.3)^2) - c_{ctrl} \tag{31}$$

where $h_{torso}$ is the height of the humanoid torso, and $c_{ctrl}$ is a control cost provided by the environment.

Given a learner observation $o_t^L$, demonstration subgoal $o_j^D$, visual backbone $\phi$, and joint predictor $f$, we let

$$d(o_t^L, o_j^D) = ||f(\phi(o_t^L)) - f(\phi(o_j^D))||_2 \tag{32}$$

This distance metric is used by ORCA to obtain $\mathcal{R}_{orca}$. Then, we compute the confidence of the learner observation embedding $c(\phi(o_t^L))$ according to 34. This results in the final reward function:

$$\mathcal{R} = c(\phi(o_t^L)) * \mathcal{R}_{orca}(o_t^L, \xi^D) + \lambda \mathcal{R}_{stability} \tag{33}$$

### B.3.3. JOINT PREDICTOR TRAINING DETAILS

**Dataset**: We collect a dataset $\mathcal{D} = \{(o_i, j_i)\}_{i=1}^N$ containing MuJoCo images $o_i$ of various poses and corresponding joint positions $j_i$. The dataset includes in total 9,038 samples. To build the dataset, we utilize a set of rollout trajectories covering a set of goal reaching tasks (such as different hand poses, doing splits, etc.), We include both successful and unsuccessful

trajectories. To ensure diversity among samples representing different stages of a trajectory, we select one frame every $k$ frames (here $k = 5$), encouraging the network to differentiate between similar images. Given the similarity of initial trajectories, we retain the first four frames only 25% of the time, and in those cases, select a random frame from a five-frame interval.

**Training**: To train our joint predictor $f \circ \phi$, we fully fine-tune a ResNet50 backbone (He et al., 2016) pre-trained on ImageNet-1K (Deng et al., 2009) with a 3-layer MLP head that projects to the joint dimension. The MLP head has layers of shape (2048, 1024), (1024, 1024), and (1024, 54), where 54 represents the number of joints (18) multiplied by the dimension per joint (3). Optimization is performed over 100 epochs using SGD with learning rate .008, batch size 16, and momentum 0.875. After training the joint predictor, we freeze the backbone weights, and train a shallow autoencoder architecture with two linear layers of shapes $(d_\phi, 32)$ and $(32, d_\phi)$ using the same parameters, where $d_\phi$ is the dimension of the backbone (2048 in this case). This provides the reconstruction loss that is used for confidence estimation, as described in B.3.4

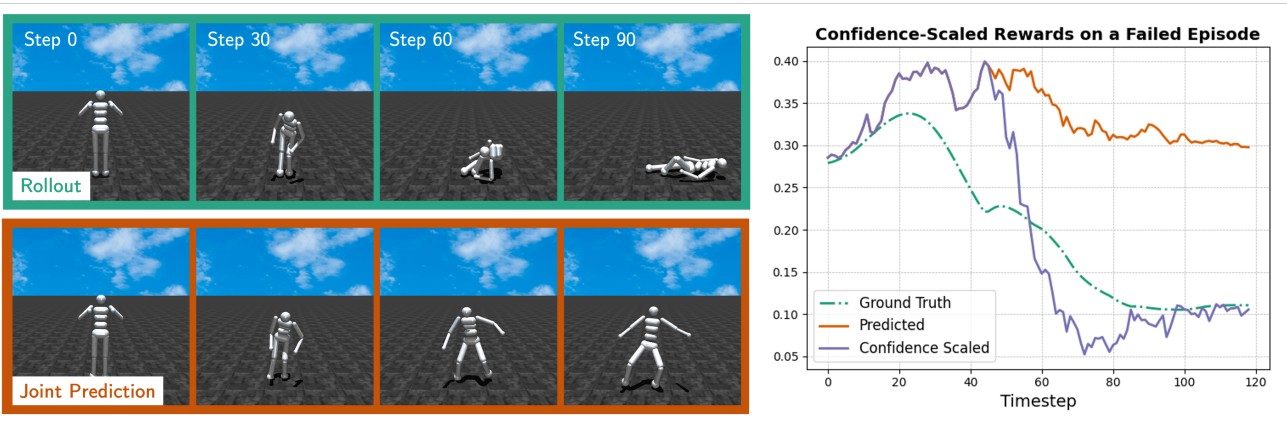

*Figure 12.* **The impact of confidence scaling on rewards for a trajectory where the robot falls down**. The ground truth trajectory is shown on top and renders of the model's joint predictions given the top frames are shown on the bottom. The rewards are computed with respect to the final reference image of the left arm wave task. Once the robot starts falling, there is a distribution shift in the images, and the model predicts that the robot is upright. This leads to high reward predictions, but also high uncertainty estimates. When corrected using the confidence function, the reward shape improves significantly.

### B.3.4. CONFIDENCE ESTIMATE AS REWARD SCALING

The use of a joint-position predictor results in an additional challenge: in an environment with unstable dynamics, there is a large space of image observations with very different joint positions, many of which are difficult to reach through robot play. During RL training, a policy can reach a state outside of $\mathcal{D}$, resulting in noisy joint predictions and rewards. To solve this problem, prior work has estimated the epistemic uncertainty of the visual model using an autoencoder architecture (Andrews et al., 2016; Frey et al., 2023). Given that the training converges, if an embedding $\mathbf{z}$ is in the domain, then the autoencoder will be able to achieve low reconstruction loss. Contrapositively, if it has high reconstruction loss on an embedding $\mathbf{z'}$, then $\mathbf{z'}$ must not be in domain.

First, we train an autoencoder to reconstruct image embeddings on the offline dataset. After training coverages, we compute the final mean $\mu_{reco}$ and standard deviation $\sigma_{reco}$ of the loss on this dataset. Then, we use a formulation inspired by Frey et al. (2023) to compute a confidence score $u(\mathbf{z_i})$:

$$c(\mathbf{z_i}) = \begin{cases} 1, & \text{if} \mathcal{L}_{reco}(\mathbf{z_i}) < \mu_{reco} \\ \exp\left(-\frac{(\mathcal{L}_{reco}(\mathbf{z_i}) - \mu_{reco})^2}{2(\sigma_{reco}k_\sigma)^2}\right), & \text{otherwise} \end{cases} \tag{34}$$

where $k_\sigma$ is a hyperparameter that controls the spread of the confidence function. In our experiments, we set $k_\sigma = 2$.

We observe that the reconstruction losses over a set of trajectories sampled from partially trained policies are skewed towards low confidence. In-domain images tend to have low reconstruction loss that is tightly clustered around the average offline loss, while out-of-domain images tend to have higher and more spread out loss. Figure 12 shows a qualitative example of

how uncertainty scaling fixes incorrect reward predictions on out-of-distribution images. The shape of the uncertainty scaled reward curve better matches the ground truth.

### B.4. RL Policy.

In the Meta-world environment, we use an identical setup and hyperparameters to Fu et al. (2024c), training the policy with DrQ-v2 over 1 million steps. We also follow Fu et al. (2024c) to repeat a policy's predicted action 2 times, which effectively shortens the episode length by half. This effect is applied both in RL training and generating demonstrations from hand-engineered policies. In MuJoCo, the policy is trained with SAC over 2 million steps, and we slightly modify the parameters from Rocamonde et al. (2024) to adapt to the shorter training period (2 million steps vs 10 million steps). All policies are trained with ground truth state observations, but we emphasize that the reward function only sees visual observations.

*Table 3.* Training hyperparameters used for experiments on both environments.

| Parameter | Meta-world (DrQ-v2) | Humanoid (SAC) |
|---|---|---|
| Total environment steps | 1,000,000 | 2,000,000 |
| Learning rate | 1e-4 | 1e-3 |
| Batch size | 512 | 256 |
| Gamma ($\gamma$) | 0.9 | 0.99 |
| Learning starts | 500 | 6000 |
| Soft update coefficient | 5e-3 | 5e-3 |
| Actor/Critic architecture | (256, 256) | (256, 256) |
| Episode length | 125 or 175 | 120 |

### B.5. Approach and Baselines.

We describe the hyperparameter details for each baseline as needed.

- `ORCA`. We use $\lambda = 1$ for temperature turn. For Meta-world tasks, we initialize the policy by loading `TemporalOT`'s checkpoint at 500k steps before training on ORCA rewards. For Humanoid tasks, we load the checkpoint at 1M steps because the total training steps is 2M.

- `OT`. We solve the entropic regularized optimal transport problem shown in (1). In Meta-world, the weight on the entropic regularization term is $\epsilon = 1$, and in Humanoid, it is $\epsilon = 0$.

- `TemporalOT`. TemporalOT (Fu et al., 2024c) was originally designed for learner and reference sequences of the same length. In this paper, we compare against a slightly stronger version of TemporalOT, which allows for a learner that is linear in the speed of the expert (either faster or slower). Specifically, we mask with a windowed diagonal matrix stretched along the longer of the learner and demonstration axis. Because TemporalOT also formulates the entropic regularized OT problem (4), the entropic weight is also $\epsilon = 1$. We use a varying mask window size, depending on the reference length. Fu et al. (2024c) used $k_m = 10$ for metaworld matched demos. We use the same for matched demos, and for mismatched demos, we let $k_m \approx \lceil \frac{|\xi|}{10} \rceil$.

- `Threshold`. This is a hand-engineered reward function with simple conditionals. `Threshold` contains two terms: the number of subgoals completed and the reward with respect to the current subgoal to follow. It initializes the subgoal to follow as the first subgoal, and it starts tracking the next subgoal, when the current subgoal's reward is above a predefined threshold. We set the threshold as $0.90$ for all Meta-world tasks and $0.70$ for all Humanoid tasks because it is a more challenging environment where it is difficult to achieve high rewards.

- `RoboCLIP`. RoboCLIP (Sontakke et al., 2024) uses a pretrained video-and-language model (Xie et al., 2018) to directly encode the video. For videos with more than 32 frames, `RoboCLIP` downsample them to 32 frames before passing them to the encoder. It defines the reward for the last timestep as the cosine similarity between the learner video's and the demonstration video's embeddings, while all previous timesteps have zero as the reward. Due to Python version issues, we train all `RoboCLIP` policies using SAC (Haarnoja et al., 2018), following the setup and code base from Sontakke et al. (2024).

### B.6. Varying Temporal Alignment Level Experiment Details.

We study the effect of how increasingly misaligned demonstrations would affect `ORCA` and `TemporalOT`'s performances. We identify two types of temporal misalignment: either the demonstration contains pauses and is slower, or the demonstration accelerates through a segment and is faster. We randomly perturb the original demonstrations of three Meta-world tasks (*Door-open*, *Window-open*, *Lever-pull*), generating 6 demonstrations per task per speed type (i.e, slow or fast).

Concretely, we first evenly split the original demonstration into five segments. To get the first three randomly perturbed demonstrations, we randomly select *one* segment and randomly change their speed. We can speed up a segment by 2, 4, 6, 8, or 10 times, and we can slow down a segment by 2, 3, 4, 5, or 6 times. To get the rest of the randomly perturbed demonstrations, we randomly select *three* segments and randomly change their speed. Then, for each task and speed type, we categorize 3 of the perturbed demonstrations as having "Low" level of misalignment and the other 3 as having "High" level of misalignment by ranking the demonstrations based on the mean absolute deviation between the segment length. The more varied the segment lengths are with respect to each other, the more temporally misaligned this demonstration is. Each perturbed demonstration is trained on a random seed, and we ensure that `ORCA` and `TemporalOT` are trained on that same seed for fair comparison.

## C. Additional Experiment Results

**Temporally Misaligned Experiment.**

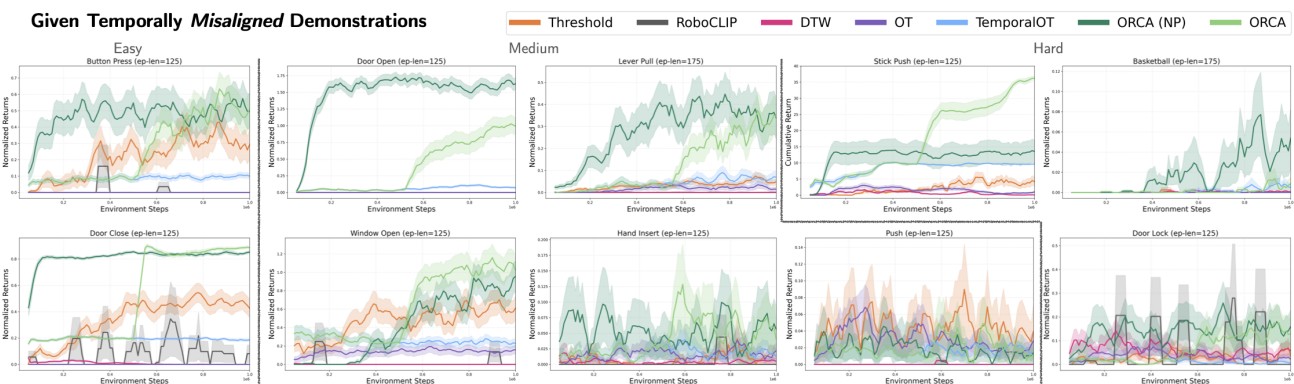

*Figure 13.* **Training Curves for All Meta-world tasks Given Temporally *Misaligned* Demonstrations** We compute the mean and standard error across the 3 training runs, each evaluated on 10 random seeds.

Fig. 13 shows training curves of all methods given temporally misaligned demonstrations. The `ORCA` plot diverges directly from `TemporalOT` at 500k steps because it is initialized with these `TemporalOT` policies, and trained for an additional 500k stops. In tasks where `TemporalOT` achieves some success by this point steps, `ORCA` performs at least as well as `ORCA(NP)`, because it can take advantage of the better initialization. In tasks like basketball and door-open, where `TemporalOT` is unsuccessful after 500k steps, `ORCA(NP)` performs better because it is trained with the `ORCA` reward for more steps. All methods fail to learn the push task, which we hypothesize is due to the visual encoder.

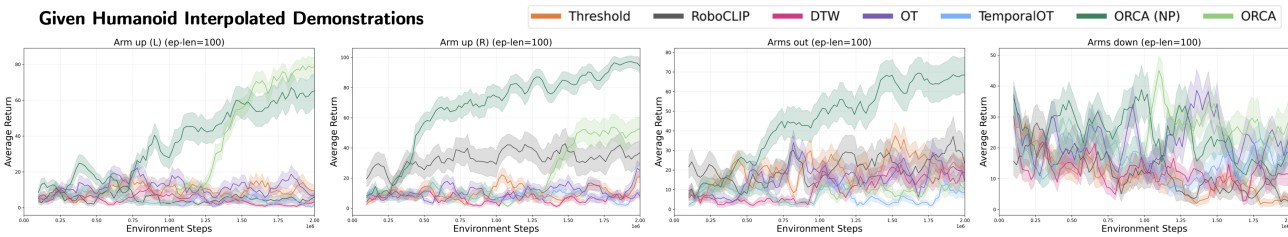

*Figure 14.* **Training curves for all humanoid tasks.** We compute the mean and standard error across the 3 training runs, each evaluated on 8 random seeds. Because `TemporalOT` achieves poor performance, `ORCA` does not benefit from pretraining, and `ORCA(NP)` is the most successful.

*Table 4.* **Performance comparison across all tasks and baselines in the Humanoid environment.** The best value within 1 standard deviation for each row is highlighted.

| Task | Threshold | RoboCLIP | DTW | OT | TOT | ORCA (NP) | **ORCA** |
|---|---|---|---|---|---|---|---|
| Arm up (L) | 7.21 (2.98) | 5.17 (2.25) | 11.38 (3.49) | 5.42 (2.75) | 5.29 (2.22) | 65.88 (8.25) | **81.62 (3.65)** |
| Arm up (R) | 13.58 (2.70) | 34.17 (7.26) | 1.12 (0.69) | 19.12 (4.10) | 7.67 (2.88) | **92.46 (4.71)** | 49.58 (5.00) |
| Arms out | 20.79 (4.47) | 7.75 (5.00) | 4.75 (2.36) | 3.75 (1.84) | 1.62 (0.75) | **72.67 (10.09)** | 8.50 (2.60) |
| Arms down | 0.00 (0.00) | 0.67 (0.56) | 3.17 (2.31) | **30.38 (7.19)** | 11.62 (3.56) | 19.71 (5.03) | **33.42 (7.20)** |
| Average | 10.40 (2.54) | 11.94 (3.77) | 5.10 (2.22) | 14.67 (3.97) | 6.55 (2.35) | **62.68 (7.02)** | 43.28 (4.61) |

Fig. 14 shows training curves of all methods given interpolated demonstrations in the humanoid environment, which are naturally temporally misaligned. Tab. 4 shows the final cumulative reward of all methods. Across all tasks, ORCA(NP) performs the best. In general, ORCA(NP) performs better than ORCA because TemporalOT achieves near 0 performance, and ORCA cannot reap the benefits of better initialization. All methods perform poorly on arms down because it is an extremely unstable position.

Fig. 15 shows a qualitative comparison between frame-level matching approaches and ORCA on the left arm up task. ORCA quickly solves the task, while the other approaches have the failure modes described in Sec. 3.

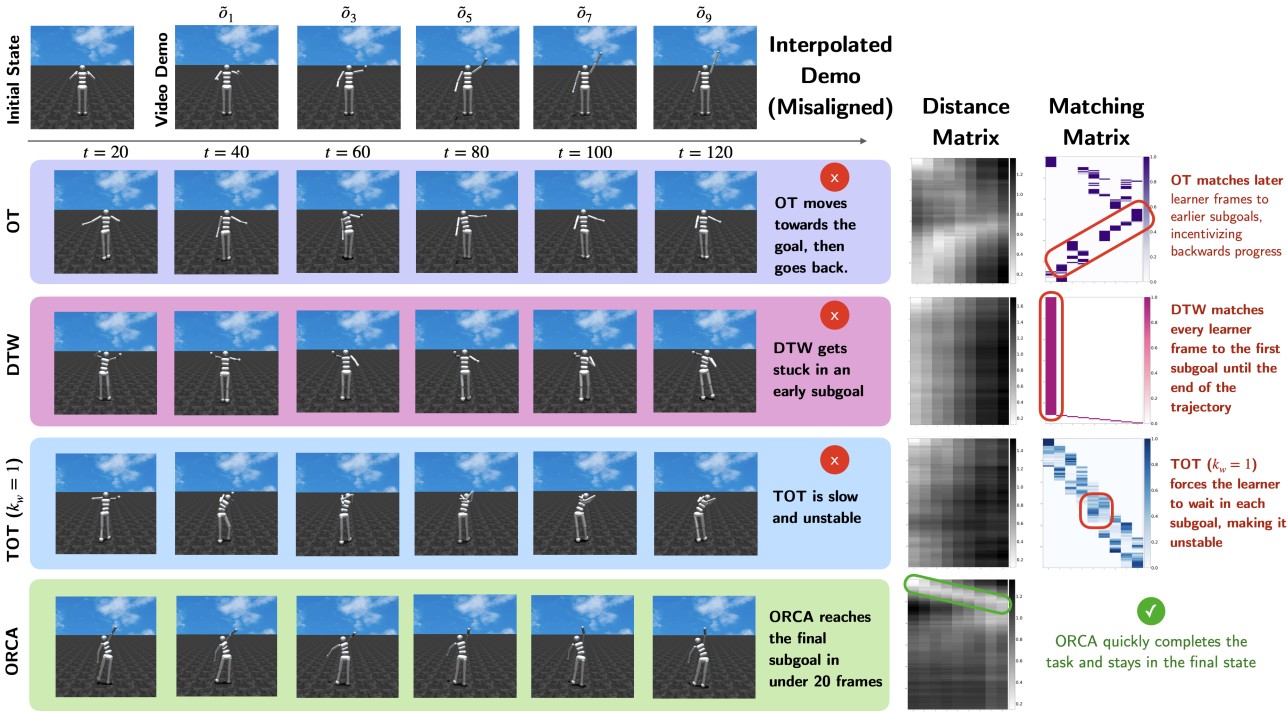

*Figure 15.* **Qualitative comparison between the frame-level matching approaches and `ORCA` when solving the Mujoco *Arm Up (L)* task**. The `ORCA` agent completes the task quickly, while the other methods exhibit the failure cases described in Sec. 3 .

**Temporally Aligned Experiment.**

Fig. 16 shows the training curves for all methods in Meta-world with temporally aligned demonstrations, and Tab. C shows the final cumulative reward of all methods. Across all tasks, ORCA performs the best. Since TemporalOT achieves good baseline performance on most of these tasks, ORCA is able to take advantage of pretraining, and outperforms ORCA(NP) on most tasks.

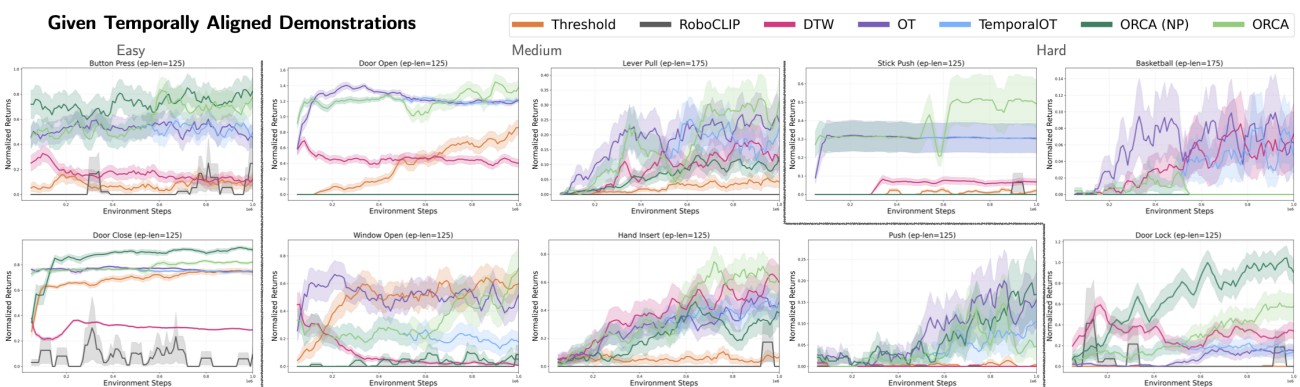

*Figure 16.* **Training Curves for All Meta-world tasks Given Temporally *Aligned* Demonstrations** We compute the mean and standard error across the 3 training runs, each evaluated on 10 random seeds.

*Table 5.* **Meta-world per-task results on temporally aligned demonstrations.** We report the mean expert-normalized return and the standard error; each task is run on three random seeds.

| Environment | RoboCLIP | Threshold | DTW | OT | TemporalOT | ORCA (NP) | **ORCA** |
|---|---|---|---|---|---|---|---|
| Button-press | 0.00 (0.00) | 0.17 (0.07) | 0.16 (0.05) | 0.45 (0.09) | 0.61 (0.09) | **0.86 (0.11)** | 0.71 (0.12) |
| Door-close | 0.47 (0.38) | 0.72 (0.02) | 0.30 (0.01) | 0.75 (0.01) | 0.74 (0.01) | **0.92 (0.01)** | 0.81 (0.01) |
| Door-open | 0.00 (0.00) | 0.91 (0.09) | 0.40 (0.05) | 1.22 (0.03) | 1.20 (0.03) | 0.00 (0.00) | **1.45 (0.11)** |
| Window-open | 0.00 (0.00) | **0.61 (0.09)** | 0.01 (0.01) | 0.61 (0.09) | 0.13 (0.06) | 0.07 (0.07) | **0.64 (0.17)** |
| Lever-pull | 0.00 (0.00) | 0.03 (0.02) | 0.14 (0.05) | 0.15 (0.06) | **0.37 (0.08)** | 0.19 (0.08) | 0.35 (0.09) |
| Hand-insert | 0.00 (0.00) | 0.11 (0.05) | 0.59 (0.11) | **0.73 (0.09)** | 0.56 (0.09) | 0.30 (0.10) | **0.64 (0.12)** |
| Push | 0.00 (0.00) | 0.00 (0.00) | 0.00 (0.00) | **0.25 (0.09)** | 0.10 (0.06) | 0.21 (0.09) | 0.12 (0.07) |
| Basketball | 0.00 (0.00) | 0.00 (0.00) | **0.11 (0.05)** | 0.06 (0.03) | 0.08 (0.08) | 0.00 (0.00) | 0.00 (0.00) |
| Stick-push | 0.00 (0.00) | 0.01 (0.01) | 0.07 (0.02) | 0.31 (0.08) | 0.30 (0.08) | 0.00 (0.00) | **0.48 (0.13)** |
| Door-lock | 0.00 (0.00) | 0.00 (0.00) | 0.30 (0.08) | 0.16 (0.06) | 0.18 (0.07) | **0.75 (0.13)** | 0.53 (0.13) |
| **Average** | 0.05 (0.05) | 0.26 (0.02) | 0.21 (0.02) | 0.47 (0.03) | 0.43 (0.03) | 0.33 (0.03) | **0.57 (0.04)** |

### C.1. Frame-Level Failure Modes in Meta-world

We show qualitative examples of the failure modes of `OT`, `TemporalOT`, and `DTW` described in Sec. 3 occuring during Meta-world policy training.

**OT and TemporalOT Fail to Enforce Ordering:** Fig. 17 shows how `OT` and `TemporalOT` with a large $k_m$ both fail to enforce subgoal ordering, thus rewarding trajectories that do not complete subgoals in the correct order.

**DTW Fails to Enforce Subgoal Coverage:** Fig. 18 shows how `DTW` fails to enforce full subgoal coverage, getting stuck in an intermediate subgoal.

### C.2. The importance of pretraining for `ORCA`

Fig. 19 shows a qualitative example of how pretraining on `TemporalOT` reward helps initialize `ORCA` in the correct basin. Without pretraining, `ORCA(NP)` gets partial credit for earlier subgoals, and gets stuck in a local minimum of immediately pushing the bottle without picking up the stick. Meanwhile, `TemporalOT` on its own moves slowly and does not pick up the stick. `ORCA` picks up the stick and completes the task quickly.

### C.3. Additional Baselines

In addition to the baselines described in the main paper, we include a classic IRL algorithm and a text-conditioned reward function. We test these baselines on all easy and medium Meta-world tasks. The results are shown in Table C.3.

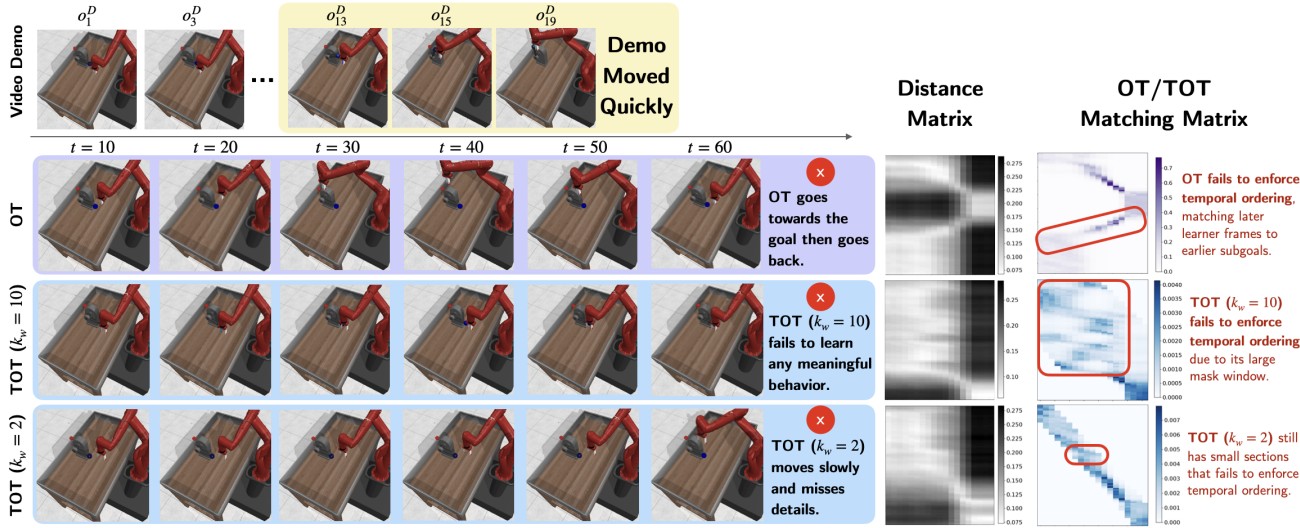

Figure 17. **Example of how `OT` and `TemporalOT` (depending on the mask window size $k_w$) fail to enforce subgoal ordering when solving a Meta-world task (*Stick-Push*).**

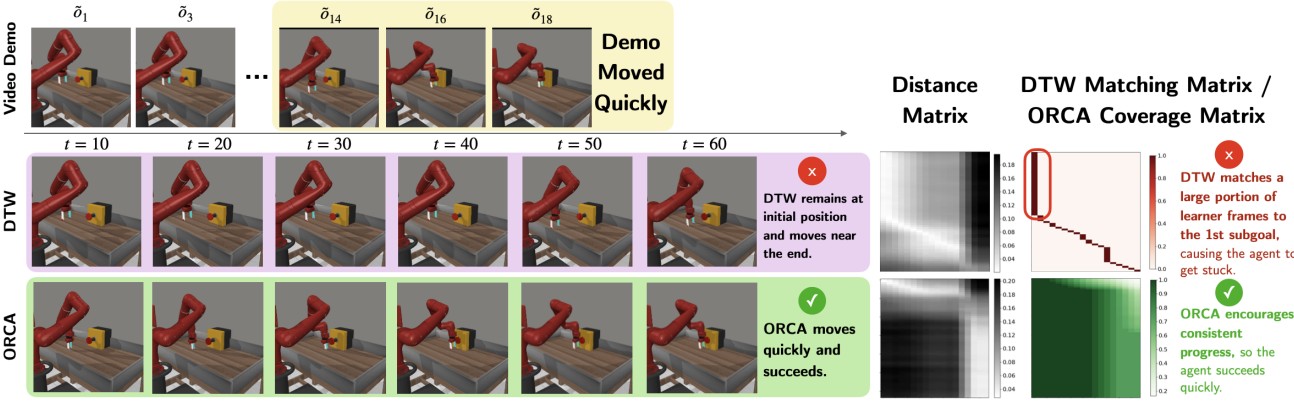

Figure 18. **Example of how `DTW` fails to enforce full subgoal coverage when solving a Meta-world task (*Lever-pull*).**

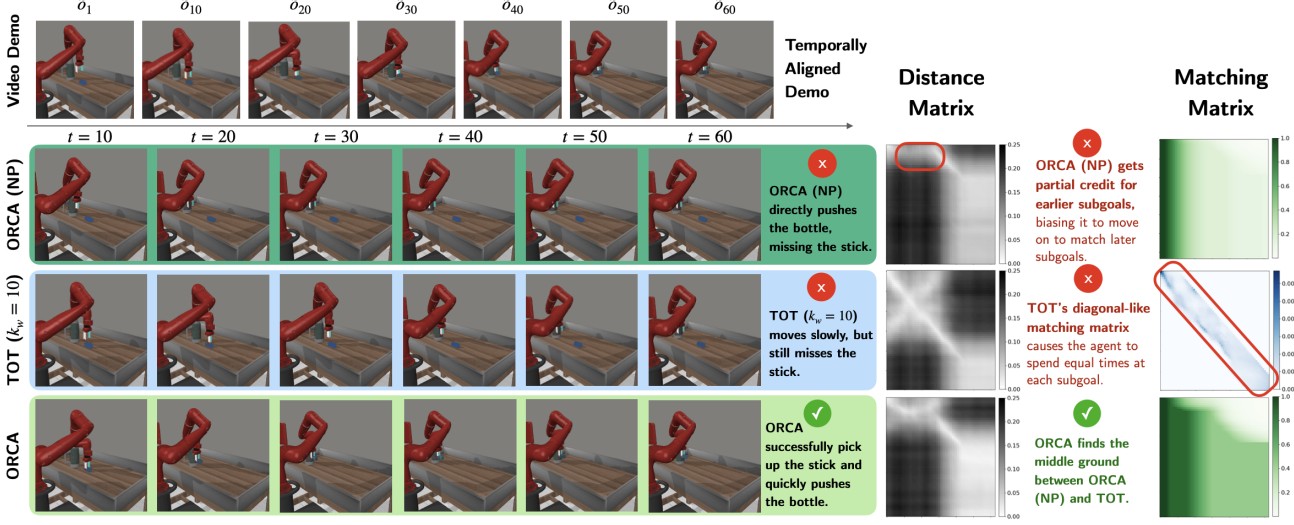

Figure 19. **Example of how pretraining improves `ORCA`'s performance.**

**GAIfO** (Torabi et al., 2019) is a traditional IRL algorithm. It jointly trains an RL policy and a discriminator between state transitions from the learner and demonstration distributions. The discriminator is used as the reward function for the policy, encouraging the agent to stay within the expert state distribution. Due to computation limitations, we used a state-based demonstration instead of a video. In contrast, ORCA and all existing baselines use video demonstrations. GAIfO outperforms TemporalOT, but it is still significantly worse than ORCA. We hypothesize that, because there is only one demonstration, GAIfO could fixate on inconsequential details instead of estimating the task reward.

**LIV** (Ma et al., 2023) is a vision language model that can be used to calculate the rewards of a video that teaches given a description of text. For this baseline, we follow the implementation of FuRL [3], which closely resembles our IRL setup. While it outperforms TemporalOT, LIV is significantly worse than ORCA. LIV uses a single text goal to describe the task, whereas ORCA is conditioned on a sequence of images, making ORCA denser and better at capturing details. Future work should explore how ORCA can be applied to sequences of text subgoals.

*Table 6.* **Meta-world per-task results on additional baselines LIV and GAIfO.** We report the mean expert-normalized return and the standard error across different methods and task difficulties.

| Category | Task | GAIfO (state-based demo) | LIV (text goal) | TemporalOT | **ORCA** |
|---|---|---|---|---|---|
| Easy | Button-press | 0.12 (0.04) | 0.00 (0.00) | 0.10 (0.02) | **0.62 (0.11)** |
| Easy | Door-close | 0.34 (0.04) | 0.34 (0.09) | 0.19 (0.01) | **0.88 (0.01)** |
| Medium | Door-open | 0.22 (0.05) | 0.00 (0.00) | 0.08 (0.01) | **0.89 (0.13)** |
| Medium | Window-open | 0.27 (0.09) | 0.72 (0.19) | 0.26 (0.05) | **0.85 (0.16)** |
| Medium | Lever-pull | 0.10 (0.03) | 0.00 (0.00) | 0.07 (0.03) | **0.28 (0.09)** |
| Total | | 0.21 (0.03) | 0.21 (0.05) | 0.14 (0.01) | **0.70 (0.05)** |

## C.4. Visual Encoder Ablation

The choice of encoder is a practical, task-dependent choice, and not the main focus of this work. In Metaworld, we followed Fu et al. (2024c) and used a pretrained Resnet50. In Humanoid, we finetuned the encoder on a set of images from the simulation environment.

We include in C.4 an ablation of ORCA with LIV (Ma et al., 2023), a robotics-specific visual encoder, and DINOv2 (Oquab et al., 2023), a standard vision model. Overall, Resnet50 achieves the best performance, although there is high variability.

*Table 7.* **Comparison of different visual backbones with ORCA.** We report the mean expert-normalized return and the standard error across different backbone architectures with their parameter counts.

| Task | ORCA+Resnet50 (26M) | ORCA+LIV (100M) | ORCA+DINOv2-L (300M) |
|---|---|---|---|
| Door-open | **1.71 (0.08)** | 1.10 (0.06) | 0.44 (0.12) |
| Window-open | 0.50 (0.14) | **1.17 (0.15)** | 0.65 (0.10) |
| Lever-pull | **0.28 (0.09)** | 0.04 (0.01) | 0.16 (0.06) |
| Total | **0.83 (0.09)** | **0.77 (0.08)** | 0.42 (0.06) |

## C.5. Image-Conditioned Policy

ORCA's goal is to estimate rewards from a single demonstration in the observation space before learning a policy, and we pose no assumptions on the policy itself. In this paper, we choose to provide the policy with ground truth states, for better comparisons with prior work. To investigate an image-conditioned policy, we trained an agent using DrQv2 [10] on a subset of the Metaworld tasks. Table C.5 shows the results. The state and image-based policies trained with ORCA have similar performance (average normalized return: $0.704 \pm 0.10 \rightarrow 0.76 \pm 0.06$). In contrast, policies trained with TemporalOT perform poorly, regardless of their input.

*Table 8.* **Comparison of policies trained on ground truth states and image observations.** We report the mean expert-normalized return and the standard error across different policy types.

| Task | TemporalOT (state) | TemporalOT (image) | ORCA (state) | ORCA (image) |
|------|--------------------|--------------------|--------------|--------------|
| Button-press | 0.10 (0.02) | 0.00 (0.00) | **0.62 (0.11)** | **0.60 (0.11)** |
| Door-close | 0.19 (0.01) | 0.12 (0.01) | **0.88 (0.01)** | **0.88 (0.02)** |
| Door-open | 0.08 (0.01) | 0.00 (0.00) | 0.89 (0.13) | **1.31 (0.12)** |
| Window-open | 0.26 (0.05) | 0.28 (0.05) | **0.85 (0.16)** | **0.79 (0.15)** |
| Lever-pull | 0.07 (0.03) | 0.00 (0.00) | **0.28 (0.09)** | **0.19 (0.08)** |
| Total | 0.14 (0.02) | 0.08 (0.01) | **0.71 (0.10)** | **0.76 (0.06)** |

## C.6. Runtime Comparison of Approaches

We experimentally tested the reward calculation latency to validate that ORCA is similar in runtime to its baselines. For each method, we evaluated 100 rollouts with a demonstration of length 100 and learner rollouts of length 100 and 300.

Table C.6 shows the average latency (ms) of different reward computation methods. ORCA is 24% faster than TemporalOT and 52% faster than RoboCLIP given rollouts of length 100, and it is comparable to threshold, OT, and DTW. LIV is the fastest method because it only considers a text goal and not a sequence of subgoals, but this leads to poor performance on most tasks. The ordering of the methods is the same for 100 and 300 frames, demonstrating that ORCA can efficiently scale to longer horizon tasks.

*Table 9.* **Latency comparison of ORCA and its baselines.** Methods are sorted by their latency in increasing order from left to right. We report the mean performance and standard error given learner rollouts of length 100 and 300.

| Learner Length | LIV | OT | Threshold | ORCA | DTW | TemporalOT | Roboclip |
|----------------|-----|-----|-----------|------|-----|------------|----------|
| 100 frames | 38.0 (4.7) | 54.0 (0.4) | 54.7 (3.8) | 56.9 (0.3) | 58.8 (0.4) | 75.1 (1.2) | 118.2 (17.9) |
| 300 frames | 130.0 (3.3) | 170.2 (0.4) | 171.0 (4.2) | 179.9 (0.6) | 185.3 (0.4) | 228.5 (2.0) | 300.2 (31.3) |

