# OpenReview forum: "Imitation Learning from a Single Temporally Misaligned Video"
_ICML.cc/2025/Conference — ICML 2025 poster_

### Official Review · Reviewer_gxJg · 2025-03-11

**Overall Recommendation:** 4

**Summary:**

The paper tackles the problem of imitation learning from a single video demonstration that may be temporally misaligned with the learner’s execution (e.g., inconsistent execution speed, pauses, etc.). The authors show that severe misalignments such as those that were created in their experiments (long pauses or x5 or x10 speedups in parts of the trajectory) can render prior imitation-learning techniques like Optimal Transport (OT), and Dynamic Time Warping (DTW) ineffective.
Specifically, they show that OT can match subgoals in the wrong order and DWT can skip goals or not complete the task in these situations..

the proposed ORdered Coverage Alignment (ORCA) computes a non-Markovian reward that measures the probability that the learner has “covered” each demonstration subgoal in the correct order. At every timestep 𝑡, the reward depends on whether the learner occupies (or closely matches) the final subgoal in the demonstration and how well it has covered all preceding subgoals.
ensuring the agent stays with the last subgoal once it is covered.

Empirically, ORCA can get stuck in local minima if the policy does not have a decent initialization. Hence, the authors first pretrain the agent with a simpler, frame-level alignment reward (specifically TemporalOT) and then switch to ORCA. This yields better coverage of all subgoals in the correct order.

Experiments are performed on a set of simulated robotic arm tasks and a set of humanoid movement-to-pose tasks.
In all experiments, intentional severe time warping is introduced.
Comparisons are performed to:
OT (Optimal Transport with entropic regularization)
TemporalOT (OT with a diagonal “mask” for partial temporal alignment)
DTW (Dynamic Time Warping, which enforces monotonic time alignment but not full coverage)
Threshold (a simpler hand-engineered approach that tracks subgoals via a distance threshold)
RoboCLIP (a transformer-based model encoding entire demonstration videos)

Results show robustness of the ORCA approach.

**Claims And Evidence:**

The claims are well supported with toy-problem demostrations of the failure modes of the other methods.

**Essential References Not Discussed:**

I am not familiat with particular studies should have been mentioned in this context.

**Experimental Designs Or Analyses:**

The experiments heightlight the strength of the method (covering all goals in the correct order when the demostration data is very warped))

**Methods And Evaluation Criteria:**

The methods that are compared against are good representatives of methods in use and halp clarify the difficulties that the proposed method can tackle.

**Other Comments Or Suggestions:**

None

**Other Strengths And Weaknesses:**

The situation in which training is based on a single example trajectory with sever timewarping seems very unnatural. In most real life situations there would be several to large amounts of demonstrations and time-warping would not be so sever (especially with tele-operated robots where robot dynamics are similar in demonstration and operation). Therefore it is hard to extrapolate from the situation tackled by this paper to the more commonly encountered use cases.

The paper does well to make use of image (as opposed to state) data in calculating the reward. It also does well to perform RL training with standard implementations using a state-based representation for the agent. In this way - the focus of the evaluation is on the fitness of the reward generating mechanism.

**Questions For Authors:**

The paper is very clear as it is.

**Relation To Broader Scientific Literature:**

The method compared against ar good candidate methods to compare against though there are certainly additional methods in this field.

**Theoretical Claims:**

The proofs in the appendix where read briefly and did not raise any red flags at the level in which they were examined.

---

> ### Author Rebuttal · Authors · 2025-04-01
>
> We are excited that the reviewer finds our claims well-supported and that our experiments highlight the strength of ORCA. Below, we address the reviewer's concern.
> # Concerns
> ## Clarification on applications of ORCA
> We clarify that our focus is on learning from temporally misaligned data. In robotics, this is a common problem even when collecting teleoperated demonstrations. Recent works handle the temporal misalignment by filtering trajectories [1,2] and manually removing pauses [3].
>
> Instead of relying on these data collection tricks, we focus on designing an algorithm that directly addresses issues of temporal misalignment. And we find the results exciting: ORCA is the most performant approach given either a temporally aligned (Fig. 4 in the paper) or a misaligned demonstration (Table 1 in the paper).
>
> ORCA can also handle multiple demonstrations by calculating the ORCA reward with respect to each demonstration and then performing a max-pool. We show that ORCA improves its performance as the number of demonstrations increases, even when the videos have different levels of temporal misalignment. For details and results, we kindly refer the reviewer to our replies to reviewer YcsA.
>
> [1] Chi et al. Universal Manipulation Interface: In-The-Wild Robot Teaching Without In-The-Wild Robots. RSS, 2024.
>
> [2] Ghosh et al. Octo: An Open-Source Generalist Robot Policy. RSS, 2024.
>
> [3] Kim et al. OpenVLA: An Open-Source Vision-Language-Action Model. CoRL, 2024.

---

### Official Review · Reviewer_xJms · 2025-03-14

**Overall Recommendation:** 3

**Summary:**

Reinforcement learning and imitation learning from a single visual demonstration is a challenging problem because the demonstration may be temporally misaligned: the demonstration and online trajectories may differ in timing. Frame-level matching is not adequate because it does not enforce the correct ordering of subgoal completion. OCRA presents a dense reward function that encourages the agent to cover all subgoals from the demonstration in the correct order. Empirical results show that ORCA significantly outperforms existing frame-level alignment methods on meta-world tasks and a simulation humanoid setup.

**Claims And Evidence:**

The three key contributions seem to be verified* and supported* (* because of the questions raised in the Methods And Evaluation Criteria and Experimental Designs Or Analyses sections).

**Essential References Not Discussed:**

In the robot learning domain, there are a few recent works that worked on learning from a few video demonstrations. It is not necessarily the case where visual imitation learning has to be framed as online RL or IRL. See below:

[1] Jain, Vidhi, Maria Attarian, Nikhil J. Joshi, Ayzaan Wahid, Danny Driess, Quan Vuong, Pannag R. Sanketi et al. "Vid2robot: End-to-end video-conditioned policy learning with cross-attention transformers." arXiv preprint arXiv:2403.12943 (2024).

[2] Fu, Letian, Huang Huang, Gaurav Datta, Lawrence Yunliang Chen, William Chung-Ho Panitch, Fangchen Liu, Hui Li, and Ken Goldberg. "In-context imitation learning via next-token prediction." arXiv preprint arXiv:2408.15980 (2024).

[3] Xu, Mengdi, Yikang Shen, Shun Zhang, Yuchen Lu, Ding Zhao, Joshua Tenenbaum, and Chuang Gan. "Prompting decision transformer for few-shot policy generalization." In international conference on machine learning, pp. 24631-24645. PMLR, 2022.

[4] Y. Duan et al., “One-shot imitation learning,” Advances in neural information processing systems, vol. 30, 2017.

[5] Di Palo, Norman, and Edward Johns. "Keypoint action tokens enable in-context imitation learning in robotics." arXiv preprint arXiv:2403.19578 (2024).

**Experimental Designs Or Analyses:**

Related to the Q3 below: based on figure 3, section 5.1, it seems that the environment for RL training is not randomized (i.e. for the door-open task, the location of the door isn’t randomized). I am not sure if this assessment is correct. The key challenge, as identified by other visual imitation learning works, is that the condition only provides a signal of the task definition, and the test time environment configuration may be quite different. It is unclear if the current experiment setup correctly accesses the model’s capability, especially since the reward function is dependent on explicit state comparisons.

**Methods And Evaluation Criteria:**

See Experimental Designs Or Analyses for the main concern. In addition, while the paper emphasizes imitation learning from a single temporally misaligned video demonstration, the policy implementation appears to rely exclusively on state-conditioned inputs, rather than directly incorporating video-based conditioning. Given the nature of the proposed ORCA reward function—which explicitly leverages visual alignment at the sequence level—it seems more natural and potentially beneficial for the policy to be conditioned jointly on both state and visual information from the demonstration.

**Other Comments Or Suggestions:**

N.A.

**Other Strengths And Weaknesses:**

N.A. See other sections for a more detailed explanation.

**Questions For Authors:**

Q1. Concerns regarding the title of the paper. Imitation learning has a special connotation (supervised learning from (visual)-proprio-action data), where the policy is not updated via online-RL. It seems to be the case that “online RL from single temporally misaligned video” seems more reasonable. If it is purely imitation learning, then a comparison with Vid2Robot [1] seems necessary.

Q2: There’s a mismatch between the demonstration and the policy learning. It is quite weird that while the demonstration only includes visual observation, the learned RL policy has access to full state information and does not take in visual observation: “All policies use state-based input, and in metaworld we include an additional feature that represents the percentage of total timesteps passed” (pg 5). This design seems counterintuitive because state information is quite hard to extract in real-world RL/robotics applications (see Vid2Robot [1], where in certain timesteps object states may be partially/not observable).

Q3: Concerns on generalization: in the setting that is being presented in the paper (figure 3, section 5.1) it seems that the environment for RL training is not randomized (i.e. for the door-open task, the location of the door isn’t randomized). Please let me know if this understanding is correct. I think the key challenge, as identified by other visual imitation learning works that are video, video-action, or state-action conditioned [1,2,3], the condition only provides a signal of the task definition, and the test time environment configuration may be quite different. It is unclear how the current version of the method can be applied to those settings.

[1] Jain, Vidhi, Maria Attarian, Nikhil J. Joshi, Ayzaan Wahid, Danny Driess, Quan Vuong, Pannag R. Sanketi et al. "Vid2robot: End-to-end video-conditioned policy learning with cross-attention transformers." arXiv preprint arXiv:2403.12943 (2024).

[2] Fu, Letian, Huang Huang, Gaurav Datta, Lawrence Yunliang Chen, William Chung-Ho Panitch, Fangchen Liu, Hui Li, and Ken Goldberg. "In-context imitation learning via next-token prediction." arXiv preprint arXiv:2408.15980 (2024).

[3] Xu, Mengdi, Yikang Shen, Shun Zhang, Yuchen Lu, Ding Zhao, Joshua Tenenbaum, and Chuang Gan. "Prompting decision transformer for few-shot policy generalization." In international conference on machine learning, pp. 24631-24645. PMLR, 2022.

**Relation To Broader Scientific Literature:**

It is broadly related to visual imitation learning, one or few-shot imitation learning literature.

**Theoretical Claims:**

The proof seems correct. However, I think the more critical questions are raised in the following section.

---

> ### Author Rebuttal · Authors · 2025-04-01
>
> We are grateful that the reviewer engages closely with our ideas. We thank the reviewer for the detailed suggestions to make our paper better, and we address the concerns below.
> # Concerns
> ## Clarification about paper title
> We agree that a portion of imitation learning literature assumes access to action data. However, imitation learning is a broad category of algorithms that includes inverse reinforcement learning. IRL first estimates "the hidden objectives of desired behavior from demonstrations" before learning a policy to maximize the expected return [1]. Our work specifically studies imitation learning from observation, where demonstrations are just visual observations [6,7]. Without action labels, IRL is a more feasible choice.
>
> Our approach closely resembles the IRL framework because the ORCA reward is estimated from demonstrations. Unlike traditional IRL methods, ORCA does not directly learn a reward function (e.g., a linear combination of features) [2,3] or a discriminator [4,5]. Instead, it is similar to works that leverage OT [6,7] to match the learner and demonstration distributions.
> ## Recent works on learning from video demonstrations
> We thank the reviewer for sharing works outside of the IRL paradigm. We will update our related works section to discuss them.
>
> We also clarify that all the mentioned works require demonstrations with action data, which ORCA and its baselines do not have access to. Notably, all of them except KAT [11] require more than 10k trajectories of observations and robot actions to train their policy model. KAT requires 10 robot trajectories with action labels as in context examples. In contrast, ORCA and baselines can estimate the rewards and train the policy with just one action-free demonstration.
> ## RL Policy Input
> ORCA's goal is to estimate rewards from a single demonstration in the observation space before learning a policy. We pose no assumptions on the policy itself. As reviewer gxJg points out, by following prior work's standard RL implementation and using a state-based policy input [7-9], we can focus on evaluating ORCA’s reward formulation.
>
> We also trained visual policies using DrQv2 [10] on a subset of the Metaworld tasks. The state and image-based policies trained with ORCA have similar performance (average normalized return: 0.704 ± 0.10 → 0.76 ± 0.06). In contrast, policies trained with TemporalOT perform poorly, regardless of their input.
>
> | Task | TemporalOT (state-based policy) | TemporalOT (image-based policy) | ORCA (state-based policy) | ORCA (image-based policy) |
> |---|---|---|---|---|
> | Button-press | 0.10 (0.02) | 0.00 (0.00) | **0.62 (0.11)** | **0.60 (0.11)** |
> | Door-close | 0.19 (0.01) | 0.12 (0.01) | **0.88 (0.01)** | **0.88 (0.02)** |
> | Door-open | 0.08 (0.01) | 0.00 (0.00) | 0.89 (0.13) | **1.31 (0.12)** |
> | Window-open | 0.26 (0.05) | 0.28 (0.05) | **0.85 (0.16)** | **0.79 (0.15)** |
> | Lever-pull | 0.07 (0.03) | 0.00 (0.00) | **0.28 (0.09)** | **0.19 (0.08)** |
> | Total | 0.14 (0.02) | 0.08 (0.01) | **0.71 (0.10)** | **0.76 (0.06)** |
> ## Clarification about environment randomization
> Following prior works [7,13], we use the default setup of Metaworld [12], which randomizes the object of interest locations on each rollout (see page 18 of [12] for exact randomization details). For Humanoid, we use the environment's default randomization scheme, where the initial positions and velocities of the joints are randomly initialized.
>
> We evaluate the RL policies on 10 randomly seeded environments, and the training environment has a different seed as well.
>
> [1 An algorithmic perspective on imitation learning. Foundations and Trends® in Robotics, 2018.
>
> [2] Apprenticeship learning via inverse reinforcement learning. ICML, 2004.
>
> [3] Maximum entropy inverse reinforcement learning. AAAI, 2008.
>
> [4] A connection between generative adversarial networks, inverse reinforcement learning, and energy-based models. 2016.
>
> [5] Generative adversarial imitation learning. NeurIPS, 2016.
>
> [6] Imitation learning from pixel observations for continuous control, 2022.
>
> [7] Robot policy learning with temporal optimal transport reward. NeurIPS, 2024.
>
> [8] Graph Inverse Reinforcement Learning from Diverse Videos. CoRL, 2023.
>
> [9] What Matters to You? Towards Visual Representation Alignment for Robot Learning. ICLR, 2024.
>
> [10] Mastering Visual Continuous Control: Improved Data-Augmented Reinforcement Learning. ICLR, 2022.
>
> [11] Keypoint action tokens enable in-context imitation learning in robotics. 2024.
>
> [12]  Meta-world: A benchmark and evaluation for multi-task and meta reinforcement learning. CoRL, 2020.

---

### Official Review · Reviewer_YcsA · 2025-03-14

**Overall Recommendation:** 4

**Summary:**

This paper tries to address the challenge of learning sequential tasks from a single visual demonstration, particularly when the demonstration is temporally misaligned with the learner's execution. This misalignment can arise from variations in timing, differences in embodiment, or inconsistencies in task execution.

The authors argue that existing imitation learning methods, which treat imitation as a frame-level distribution-matching problem, fail to enforce the correct temporal ordering of subgoals or ensure consistent progress. They propose a novel approach ORCA that defines matching at the sequence level.

The key idea is that successful matching occurs when one sequence covers all subgoals in the same order as the other sequence. The ORCA reward function is recursively defined, considering both the learner's current state and whether it has covered previous subgoals correctly.

Experiments on Meta-world and Humanoid-v4 tasks demonstrate that agents trained with the ORCA reward significantly outperform existing frame-level matching algorithms.

## update after rebuttal
I still recommend acceptance since I don't have major concerns regarding this paper.

**Claims And Evidence:**

I think most of the claims in this submission are supported by evidence.

**Essential References Not Discussed:**

N/A

**Experimental Designs Or Analyses:**

I went through all the experiments, and most of them make sense to me.

**Methods And Evaluation Criteria:**

Yes, they make sense to me.

**Other Comments Or Suggestions:**

- The formatting at line 141 is very weird.

**Other Strengths And Weaknesses:**

### Strengths

- I appreciate the structure of this paper. It begins by explaining why OT and DTW are insufficient, followed by introducing the proposed method. The examples are intuitive and effectively demonstrate scenarios where OT and DTW fail. This paper is highly engaging to read.
- The key insight of this paper is clear and intuitive: matching should be defined at the sequence level instead of  the frame level when learning rewards for sequence-matching tasks. This idea is highly logical and resonates well.
- The paper provides a theoretical analysis of both the limitations of existing frame-level matching approaches (OT, DTW, TemporalOT) and the properties of ORCA. Though the proofs are actually quite simple and straightforward, they still strengthen the paper's claims.


### Weaknesses

- The abstract and introduction mention variations in embodiment, but no experiments are conducted to evaluate performance under this setting.
- As noted in Section 4, the ORCA reward is non-Markovian, which violates some assumptions in standard RL settings (MDP). It remains unclear if this could make RL training harder.
- Compared to learning from a single temporally misaligned video demonstration, a more common scenario involves multiple demonstrations for a task. Extending the proposed ORCA reward to handle multiple demonstrations appears non-trivial.

**Questions For Authors:**

- The tasks used in the experiments are from Meta-World and MuJoCo, which include many tasks that are not utilized in this paper. Could the authors clarify the rationale for excluding those tasks? Is it because ORCA does not perform well on them? This is not a criticism, as I understand that no method excels across all tasks.

**Relation To Broader Scientific Literature:**

Prior inverse RL methods, especially those leveraging optimal transport primarily focus on frame-level matching. These methods often rely on Markovian distance metrics. Other works use Dynamic Time Warping to align trajectories temporally. ORCA shifts the focus from frame-level to sequence-level matching.  This is important for tasks where the order of subgoals is critical.

**Theoretical Claims:**

I briefly checked the proofs in Appendix A.1 and did not identify any significant errors.

---

> ### Author Rebuttal · Authors · 2025-04-01
>
> We thank the reviewer for finding our work clear, intuitive, and engaging. Thank you for the feedback that helps further strengthen it. We will update the paper to fix minor formatting issues, and we address the reviewer's questions and concerns below.
> # Questions
> ## Q1: "Could the authors clarify the rationale for the chosen tasks?"
> For Metaworld, we evaluated ORCA and baselines on all nine tasks shown in TemporalOT [1], the most closely related work. We also added one more task, Door-close, because originally there was only one task (Button-press) in the Easy category.
>
> To evaluate ORCA's effectiveness on difficult control tasks, we include the MuJoCo Humanoid Environment, following the setup in prior work [2]. Because the original tasks (walking, standing up) lack demonstrations, we define tasks where we can generate demonstrations via interpolation from the initial to the goal joint state.
> # Concerns
> ## Multiple Demonstration Videos
> Same as prior work TemporalOT [1], ORCA can handle multiple videos by calculating the ORCA reward with respect to each demonstration and max-pooling to obtain the final reward.
>
> To study how the number of demonstrations affects performance, we ran TemporalOT and ORCA with 4, 10, and 20 temporally misaligned videos on a subset of Metaworld tasks (Door-open, Window-open, Lever-pull). These demonstrations have variations due to randomly initialized object locations, but they have the same speed. The table below reports the average normalized return with standard error.
>
> As the number of demonstrations increases, ORCA's performance also improves. In contrast, although TemporalOT handles multiple videos in the same way and benefits from having more than one demonstration, its performance soon starts decreasing as the number of demonstrations further increases.
>
> | # Demos	| 1           	| 4           	| 10          	| 20          	|
> |------------|-----------------|-----------------|-----------------|-----------------|
> | TemporalOT | 0.14 (0.02) 	| 0.36 (0.02) 	| 0.30 (0.02) 	| 0.25 (0.02) 	|
> | **ORCA**   | **0.67 (0.08)** 	| **1.02 (0.07)** | **1.08 (0.07)** | **1.10 (0.08)** |
>
> In addition, per reviewer SsGe's request, we study how multiple videos of different speeds affect ORCA performance. For each task, we randomly sample one video demonstration from each category of temporal misalignment Slow (H), Slow (L), Fast (L), Fast (H), as shown in Fig. 4. Policies trained with the ORCA reward and 4 demonstrations with different speed are still able to improve performance compared to having only 1 video (0.67 -> 0.77), although demonstration videos with different speeds have slightly worse performance than videos with the same speed.
>
> | # Demos	| 1           	| 4 (Different Speed) | 4 (Same Speed)  |
> |------------|-----------------|---------------------|-----------------|
> | TemporalOT | 0.14 (0.02) 	| 0.16 (0.02)     	| 0.36 (0.02) 	|
> | **ORCA**   | **0.67 (0.08)** 	| **0.77 (0.08)** 	| **1.02 (0.07)** |
> ## RL Training for Non-Markovian Tasks
> We study sequence-matching tasks where it is critical to follow the entire sequence in the correct order, thereby making the true task objective non-Markovian. ORCA and all baselines face this challenge during RL training.
>
> Prior works have explored learning policies conditioned on a belief or sequential context [3,4], and we will include these in our related work. We found it sufficient to add an additional feature to the policy input that represents the percentage of total timesteps passed. We leave exploring policy input spaces as interesting future works.
> ## Clarification about applications of ORCA
> We clarify that our focus is on learning from temporally misaligned data. This is a common problem in teleoperation data with the same embodiment, and we mention the cross embodiment setting simply to provide more context. Recent works handle temporal misalignment in teleoperation data by filtering trajectories [5,6] and manually removing pauses [7]. In contrast, we focus on designing an algorithm that can directly address issues of temporal misalignment without relying on post-processing tricks.
>
> [1] Fu et al. Robot policy learning with temporal optimal transport reward. NeurIPS, 2024.
>
> [2] Rocamonde et al. Vision-Language Models are Zero-Shot Reward Models for Reinforcement Learning. ICLR, 2024.
>
> [3] Igl et al. Deep Variational Reinforcement Learning for POMDPs. ICML, 2018.
>
> [4] Qin et al. Learning non-Markovian Decision-Making from State-only Sequences. NeurIPS, 2023.
>
> [5] Chi et al. Universal Manipulation Interface: In-The-Wild Robot Teaching Without In-The-Wild Robots. RSS, 2024.
>
> [6] Ghosh et al. Octo: An Open-Source Generalist Robot Policy. RSS, 2024.
>
> [7] Kim et al. OpenVLA: An Open-Source Vision-Language-Action Model. CoRL, 2025.

---

> > ### Comment · Reviewer_YcsA · 2025-04-05
> >
> > Thank you to the authors for their response. I still recommend acceptance.

---

### Official Review · Reviewer_HynH · 2025-03-14

**Overall Recommendation:** 3

**Summary:**

This paper studies how to provide a policy with rewards using a single video. The paper argues that temporal misalignment may occur due to pauses in the video. The authors design an algorithm that calculates the probability that the policy's frame at time t corresponds to the video's frame at time j, thereby providing better rewards. Experiments on MetaWorld validate the effectiveness of the proposed method.

**Claims And Evidence:**

For the results of Metaworld, it seems that none of the baseline methods are effective, and the significant improvement of OCRA appears to be due to the weakness of the baselines.

**Essential References Not Discussed:**

Some works that generate rewards based on text-video pairs rather than a single video have not been discussed. In comparison, such works seem to be more scalable.

LIV: Language-Image Representations and Rewards for Robotic Control

**Experimental Designs Or Analyses:**

I have reviewed Table 1 and Table 2, and both seem to have a significant advantage over the baseline ToT. However, I am not sure whether the reward proposed in the paper can be widely applied.

**Methods And Evaluation Criteria:**

In many cases, we have more than one video. The article can be extended to multiple events and situations and compared with a better baseline, such as LIV: Language-Image Representations and Rewards for Robotic Control.

**Other Comments Or Suggestions:**

No

**Other Strengths And Weaknesses:**

Strengths
1. The motivation of the paper is well-developed, with thorough and thoughtful reasoning.

Weaknesses
1. The experiments in the paper are not sufficiently comprehensive. I would expect to see the application of the reward model in more realistic scenarios. For example, in the Libero benchmark, can the reward model be used to perform rl to enhance a pre-trained diffusion policy?
You can refer to PARL for preforming rl for diffusion policy.

PARL: A Unified Framework for Policy Alignment in Reinforcement Learning from Human Feedback

**Questions For Authors:**

This method does not seem to require training. Is it scalable? How does it handle multiple videos? Will having more videos enhance the effect?

How is the reward calculation time? For very long videos exceeding 300 frames, will computing the distance matrix each time be very time-consuming? How much time does it take?

**Relation To Broader Scientific Literature:**

The paper may propose a highly general reward function that can provide rewards based on a video.

**Theoretical Claims:**

No theory in this paper.

---

> ### Author Rebuttal · Authors · 2025-04-01
>
> We appreciate the reviewer's valuable feedback on how we can strengthen our work. We answer the questions and address the concerns below.
> # Questions
> ## Q: “Can ORCA improve with multiple videos?”
> ORCA's performance improves with more demonstration videos. We kindly refer the reviewer to our reply to reviewer YcsA for the experiment setup and detailed results.
> ## Q: “What is the reward calculation time?”
> ORCA is significantly faster than TemporalOT and Roboclip, and is comparable to all other frame-level matching baselines, even for very long videos. A detailed analysis of ORCA’s runtime is in our reply to reviewer SsGe.
> ## Q: "Can ORCA enhance a pre-trained diffusion policy via RL?"
> Yes, ORCA rewards can be applied to any pretrained policy, as shown by its effectiveness at improving policies pretrained with TemporalOT. Prior work [1] has explored using an OT reward to finetune a behavior cloning policy, and ORCA can easily be extended to this framework. However, as we focus on evaluating different reward formulations, we leave different pretraining strategies as interesting future work.
> # Concerns
> ## Clarification on existing baselines' performance
> Our baselines have high normalized return given temporally aligned demonstrations (OT: 0.47, TemporalOT: 0.43). However, these baselines degrade when the demonstration is misaligned. In contrast, ORCA performs the best on both aligned (0.57) and misaligned (0.50) demonstrations. We include additional baselines below, which are more competitive under misalignment.
> ## Additional Baselines
> We emphasize that we focus on learning a policy from a single video demonstration, and we consider text-based guidance to be an interesting problem for future work. Based on the reviewer’s suggestions, we introduce two new baselines.
>
> ### Text-Conditioned Reward Formulation: LIV [2]
> LIV is a vision-language model that can be used to calculate rewards for a learner video given a text description. For this baseline, we follow the implementation of FuRL [3], which closely resembles our IRL setup. The table below shows the average normalized return of LIV on all easy and medium Metaworld tasks.
>
> While outperforming TemporalOT, LIV performs significantly worse compared to ORCA. LIV uses a single text goal to describe the task, whereas ORCA is conditioned on a sequence of images, making ORCA denser and better at capturing details.
>
> | Category | Task | GAIfO [4] (state-based obs)  | LIV [2] (same setup as [3]) | TemporalOT | **ORCA** |
> |---|---|---|---|---|---|
> | Easy | Button-press | 0.12 (0.04) | 0.00 (0.00) | 0.10 (0.02) | **0.62 (0.11)** |
> | Easy | Door-close | 0.34 (0.04) | 0.34 (0.09) | 0.19 (0.01) | **0.88 (0.01)** |
> | Medium | Door-open | 0.22 (0.05) | 0.00 (0.00) | 0.08 (0.01) | **0.89 (0.13)** |
> | Medium | Window-open | 0.27 (0.09) | 0.72 (0.19) | 0.26 (0.05) | **0.85 (0.16)** |
> | Medium | Lever-pull | 0.10 (0.03) | 0.00 (0.00) | 0.07 (0.03) | **0.28 (0.09)** |
> |  | Total | 0.21 (0.03) | 0.21 (0.05) | 0.14 (0.01) | **0.70 (0.05)** |
>
> We also point out ORCA's strength: it works with any visual encoder, including LIV. In our reply to reviewer SsGe, we evaluate ORCA with LIV as the encoder.
>
> ### IRL Baseline: GAIfO [4]
> We include GAIfO, a classical IRL algorithm. It trains a discriminator, alongside the policy, to differentiate state transitions between the learner and demonstration distributions and uses it as the reward function. Due to compute limitations, we use a state-based demonstration instead of a video. In contrast, ORCA and all existing baselines use video demonstrations.
>
> While GAIfO is a competitive baseline (outperforming TemporalOT), it still performs significantly worse compared to ORCA. We hypothesize that, because there is only one demonstration, GAIfO could fixate on inconsequential details instead of estimating the task reward.
> ## The experiments are not sufficiently comprehensive
> Prior works use Metaworld for all experiments [3, 5, 6], and we test all of the tasks in [5], the most closely related work. We additionally include 4 more difficult control tasks in the Humanoid environment to demonstrate the viability of ORCA in a variety of environments and conditions.
> ## Related works that learn from text-video pairs
> We thank the reviewer for pointing out related work in learning from text-video pairs, and we will update the related work to include LIV, as well as other video learning methods suggested by reviewer xJms.
>
> [1] Watch and match: Supercharging imitation with regularized optimal
> transport. CoRL, 2024.
>
> [2] LIV: Language-image representations and rewards for robotic control. ICML, 2023.
>
> [3] FuRL: Visual-Language Models as Fuzzy Rewards for Reinforcement Learning. ICML, 2024.
>
> [4] Generative Adversarial Imitation from Observation. ICML, 2019.
>
> [5] Robot policy learning with temporal optimal transport reward. NeurIPS, 2024.
>
> [6] Imitation learning from observation with automatic discount scheduling. ICLR, 2024.

---

### Official Review · Reviewer_SsGe · 2025-03-16

**Overall Recommendation:** 3

**Summary:**

This paper focuses on learning sequential tasks from a single temporally misaligned video, which belongs to the imitation learning paradigm. They propose a novel reward function - ORCA, which measures the matching at the sequence level to ensure that the agent covers all subgoals in the correct order. Experiments show that compared with the best frame by frame-level method, ORCA outperforms in several tasks from Meta-world and Humanoid.

**Claims And Evidence:**

clear

**Essential References Not Discussed:**

no

**Experimental Designs Or Analyses:**

yes

**Methods And Evaluation Criteria:**

yes

**Other Comments Or Suggestions:**

see Strengths And Weaknesses

**Other Strengths And Weaknesses:**

Paper Strengths
	This paper proposes a novel reward function in which it provides rewards by calculating the probability that the agent covers all subgoals in the correct order, enhancing the performance of imitation learning in a time-shifted demonstration.
	Through rigorous mathematical proof, this paper shows that ORCA can overcome the failure cases of traditional methods in subgoal ordering and coverage.
	This paper achieved superior results on different tasks and the selection of tasks is sufficient(Easy-Medium-Hard).
Paper Weaknesses
	The effectiveness of ORCA can be further clarified. ORCA is a dense reward function and implemented by dynamic programming. Thus, how about the training wall-clock time of ORCA? Is it time-consuming?
	The importance of pretraining is confusing. In Sec. 4.3, the authors explain that they pretrain the agent with TemporalOT and then train with the proposed ORCA. However, Table1 and Table 2 show that ORCA without pretraining might outperform ORCA in some tasks, which is inconsistent. Does this mean that ORCA has higher quality requirements for pretraining? It's necessary to analyze when it's better to introduce pretraining, otherwise it may indicate that ORCA would be difficult to deploy in real-world.
	The acquisition of the distance function d(∙,∙) can be further explored. In the paper, the authors utilize ResNet50 to extract visual features and the compute cosine similarity. In Fig. 10, it's found that off-the-shelf visual encoders produce noisy rewards in Mujoco and they clarify that this may require further online fine-tuning. Will different structures or scales of visual encoders (such as CLIP, R3M or MOCO) have an impact on ORCA in Meta-world or Humanoid? Please refer to R1 for details.
R1: Hu Y, Wang R, Li L E, et al. For pre-trained vision models in motor control, not all policy learning methods are created equal[C]//International Conference on Machine Learning. PMLR, 2023: 13628-13651.
	The applicability of ORCA may need further clarification. In Fig. 4, the authors show that ORCA performs worsen when the demonstrations are slowed down and longer than the learner trajectory. Does this mean that slow demonstrations would affect the quality of skill learning and lead to slow speed during inference?
	ORCA could reach SOTA with only one video. So, will more demonstrations lead to better results? As mentioned in R2, the proposed method fails when multiple demonstrations are given. Thus, when more demonstrations are provided, especially when the speeds of different demonstrations are different (slow and fast), will the performance of ORCA be affected?
R2: Sontakke S, Zhang J, Arnold S, et al. Roboclip: One demonstration is enough to learn robot policies[J]. Advances in Neural Information Processing Systems, 2023, 36: 55681-55693.
The writing can be improved. For example, the reference of 'Table 4.3' in Line 329 and 'Table 2' in Line 311 should be unified.

**Questions For Authors:**

see Strengths And Weaknesses

**Relation To Broader Scientific Literature:**

none

**Theoretical Claims:**

yes

---

> ### Author Rebuttal · Authors · 2025-04-01
>
> We are grateful for the reviewer’s deep engagement with our paper, and we thank them for feedback that strengthens its clarity. We will update the paper to unify our references to tables. Please find below our responses to questions and concerns.
> # Questions
> ## Q: "When should pre-training be performed with ORCA?"
> We agree that there is a balance between initializing the policy in a good basin via pretraining and directly optimizing ORCA rewards. We recommend using pretraining because it results in the best and most consistent performance.
>
> Given a temporally misaligned demonstration, ORCA outperforms all baselines on all Metaworld and Humanoid tasks, not including the ablation ORCA (No Pretraining). Additionally, given a temporally aligned demonstration, ORCA still consistently outperforms baselines on 8/10 tasks, not including ORCA (No Pretraining). The 2 tasks that ORCA does not work well on (push and basketball) are difficult for all methods. Overall, ORCA is a consistent choice for good performance, given both temporally misaligned and aligned demonstrations.
> ## Q: "Will more demonstrations lead to better results? What happens when different demonstrations have different speeds?"
> We show that having access to more demonstrations leads to better ORCA performance, even when these demonstrations have significantly different speeds. We kindly refer the reviewer to our reply to reviewer YcsA for the experiment setup and detailed results.
> ## Q: "What's the training wall-clock time of ORCA?"
> Empirically, the latency of ORCA reward calculation is significantly less than both TemporalOT [2] and RoboCLIP [3], and is almost identical to optimal transport, DTW, and threshold. The table below shows the average latency (ms) of different reward computation methods. We evaluated 100 rollouts with a demonstration of length 100 and learner rollouts of length 100 and 300, respectively. Experiments were performed on an NVIDIA RTX A6000 GPU.
>
> ORCA is 24% faster than TemporalOT and 52% faster than RoboCLIP given rollouts of length 100, and it is comparable to threshold, OT, and DTW. LIV is the fastest method because it does not consider demonstration frames, but this leads to poor performance on most tasks.
>
> | method | LIV | OT | Threshold | ORCA | DTW | TemporalOT | Roboclip |
> |---|---|---|---|---|---|---|---|
> | 100 rollout frames | 38.03 ± 4.69 | 54.03 ± 0.35 | 54.69 ± 3.81 | 56.93 ± 0.33 | 58.82 ± 0.37 | 75.12 ± 1.15 | 118.21 ± 17.94 |
> | 300 rollout frames | 129.96 ± 3.29 | 170.17 ± 0.44 | 170.95 ± 4.16 | 179.85 ± 0.62 | 185.27 ± 0.42 | 228.46 ± 1.95 | 300.17 ± 31.30 |
>
> ORCA’s time complexity is the learner rollout horizon multiplied by the length of the demonstration, which is also the lower bound time complexity of any method that relies on a frame-level distance matrix (including all OT-based methods).
> ## Q: "How does ORCA perform with different visual encoders?"
> The choice of encoder is a practical, task-dependent choice, and not the main focus of this work. In Metaworld, we followed prior work [2] and used a pretrained Resnet50. In Humanoid, we finetuned the encoder on a set of images from the simulation environment.
>
> We include below an additional ablation of ORCA with LIV [4], a robotics-specific visual encoder, and DINOv2 [5], a standard vision model. Overall, Resnet50 achieves the best performance, although there is high variability. This substantiates the observations in [1]: "evaluation [of visual encoders] based on RL methods is highly variable."
>
> | Task | ORCA+Resnet50 (26M) | ORCA+LIV (100M) | ORCA+DINOv2-L (300M) |
> |---|---|---|---|
> | Door-open | **1.71 (0.08)** | 1.10 (0.06) | 0.44 (0.12) |
> | Window-open | 0.50 (0.14) | **1.17 (0.15)** | 0.65 (0.10) |
> | Lever-pull | **0.28 (0.09)** | 0.04 (0.01) | 0.16 (0.06) |
> | Total | **0.83 (0.09)** | **0.77 (0.08)** | 0.42 (0.06) |
> ## Q: "Do slow demonstrations affect learning and lead to a slow policy?"
> Given slow demonstrations, TemporalOT performs poorly because it distributes the assignment over more frames. Thus, it misses important details, which causes ORCA to have similar failure modes because it is initialized in a worse basin.
>
> Nevertheless, there are no temporal constraints in the ORCA formulation, so ORCA policies trained on slow demonstrations complete tasks at a similar speed compared to fast demonstrations ([as shown in the figure from an anonymized link](https://raw.githubusercontent.com/orcaicml/ICML_Rebuttal_2025/refs/heads/main/image.png)).
>
> [1] For pre-trained vision models in motor control, not all policy learning methods are created equal. ICML, 2023.
>
> [2] Robot policy learning with temporal optimal transport reward. NeurIPS, 2024.
>
> [3] Roboclip: One demonstration is enough to learn robot policies. NeurIPS, 2023.
>
> [4] LIV: Language-image representations and rewards for robotic control. 2023.
>
> [5] DINOv2: Learning Robust Visual Features without Supervision. TMLR, 2024.

---

### Decision · Program_Chairs · 2025-05-01

**Decision:**

Accept (poster)

**Comment:**

The paper proposes a novel way to compute a "tracking score" for imitating a (possibly) temporally misaligned demonstration (e.g., a video).

There is general consensus among the reviewers about the importance of the topic, the soundness of the proposed methods, and its empirical effectiveness. For this reason I propose acceptance for the paper.

I strongly encourage the authors to integrate the discussion from the rebuttal in the final revision of the paper, in particular the new comparisons and the discussion on language-vision models